# Concert: Context-Aware Randomized Smoothing via Colorization-Based Entropy

## Abstract

Randomized smoothing provides neural network models with verifiable robustness against adversarial attacks. Most randomized smoothing defenses certify a probabilistic guarantee within an $\ell_p$ norm ball around each data point. However, certifying sufficient robustness radii towards perturbations that are large in $\ell_p$ norm requires smoothing the model input using a large variance. Consequently, the resulting classifier usually exhibits poor certified accuracy. Moreover, it might be impossible to obtain large robustness radii for certain data points and models despite a large-variance smoothing noise being adopted. For instance, these robustness guarantees are often vulnerable towards unrestricted or semantic perturbations, which have large $\ell_p$ norm but still remain imperceptible to human eyes. In this paper, we propose CONCERT, a method for certifying the robustness of neural network-based image classifiers with context-aware smoothing noise under the guidance of pixel-wise entropy values measured by a colorization model. In lieu of sampling noise from univariate Gaussian distributions, we increase the Gaussian variance in high-entropy dimensions that are more vulnerable to adversarial manipulations, while keeping a moderate variance in low-entropy dimensions. CONCERT acquires larger robustness radii on input dimensions that are prone to adversarial perturbations while preserving certified accuracy since other input dimensions are not significantly randomized. We show that our method's certified accuracy and robustness radius on benign images are on par with that of the state-of-the-art smoothing techniques while outperforming them on semantically perturbed images.

## 1 Introduction

The robustness guarantee towards adversarial attacks has been a sought-after property of neural networks while deploying secure and trustworthy machine learning applications. However, achieving robustness is not trivial due to the fast-paced contest between adversaries and defenders. Compared to empirical defenses, certified defenses provide a more sound security guarantee on the robustness of the model against adversarial perturbations since they can provably defeat adversarial examples in the vicinity of benign samples. Most of the existing certified defenses provide a symmetric robustness guarantee for each data point, *i.e.,* the defenses are dimension-agnostic by considering a robust region in $\ell_p$ norm. As a ramification, the robust region of each data sample is limited by the *worst dimension*, Although higher radii can be obtained on other dimensions. Furthermore, the defenses may compromise the utility of the protected model on benign data that is in the *vicinity* of decision boundaries (see Figure 1a).

Current randomized smoothing methods randomize the model prediction by adding random additive noise (*i.e.,* smoothing noise) to the model input. In general, it is challenging to simultaneously achieve both a high certified accuracy and a large robustness radius. The magnitude of the smoothing noise is limited in order to maintain a reasonable certified accuracy, which results in statistically small robustness radii while certifying the inputs of the smoothed classifier. This utility-security trade-off implies that defenses can be circumvented by adversarial perturbations that are substantial in $\ell_p$ norm, yet remain imperceptible. For instance, a colorization-based attack can escape from the $\ell_p$ defenses when the norm of the perturbation is greater than the radius of the robustness region (Bhattad et al., 2019). Moreover, the smoothing distribution operates independently from the base classifier. This property prohibits further customization of the smoothing

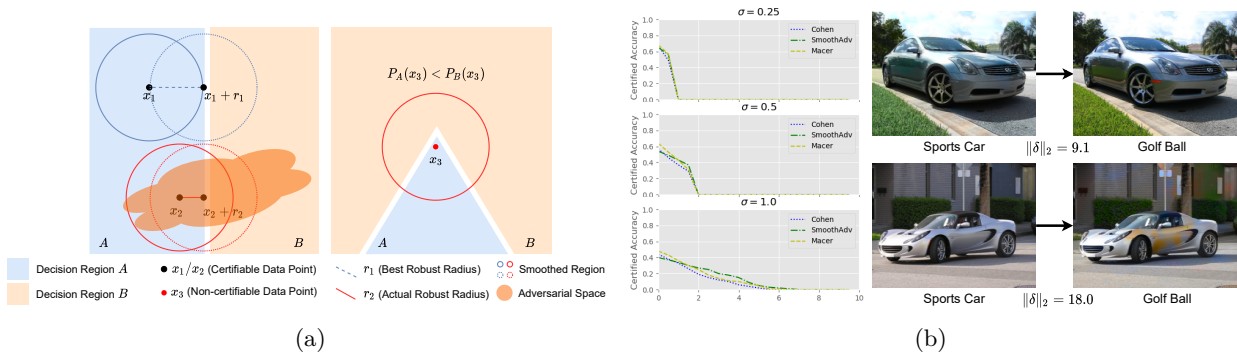

(a)                      (b)

Figure 1: (a) The limitation of symmetric (*e.g.,* $\ell_p$) robustness certificate. Consider an $\ell_2$ certificate in a two-dimensional decision region $A$ that is linearly separated from region $B$. The actual robust radius falls short of the best robust radius obtained by the smoothing noise, due to the ***worst dimension*** (horizontal). Moreover, some data points are naturally ***non-certifiable*** in decision regions such as the one depicted in Figure 1a (right), the second figure, which incurs utility loss. Importantly, the certificate is not tight for asymmetric adversarial spaces. (b) The trade-offs limit the radius that can be certified. As a consequence, current defenses are susceptible to unrestricted perturbations whose $\ell_2$ surpasses the largest certifiable radius.

distribution based on the classifier's output. Some recent work aims to address this problem by learning sample-dependent noise (Alfarra et al., 2022; Wang et al., 2020; Chen et al., 2021) for $\ell_p$ certification. However, these methods rely on per-sample calibration, which is computationally expensive. These limitations with current methods are illustrated in Figure 1a.

To address the aforementioned problems, we propose CONtext-aware robustness CERTIfication (CONCERT), a certifiable defense against unrestricted adversaries. This defense is designed to address the difficulty in quantifying the perturbation space of unrestricted adversaries, which can make certifying robustness against these types of attacks challenging. CONCERT models the smoothing distribution as a distribution conditioned on the input data and the decision region of the base classifier. A colorization model trained on ImageNet is used to estimate pixel-wise entropy values, which are then used to regularize our training objective and, therefore, help refine the variance of the smoothing distribution. The intuition behind this is that large perturbations are less noticeable and thus more likely to occur on high-entropy pixels, since the pixel values are naturally ambiguous and unpredictable. Furthermore, by increasing the noise variance on high-entropy dimensions, the certification process allocates higher robustness radii to vulnerable pixels in the input. Our experimental results show that CONCERT obtains on-par performance with state-of-the-art randomized smoothing techniques in the worst case. Moreover, CONCERT can better invalidate unrestricted perturbations, while many existing methods are susceptible to such semantic perturbation attacks.

## 2 Background and related work

*Randomized smoothing* (RS) was first introduced to provide a certifiable robustness guarantee to adversarial perturbations, along with *differential privacy* (Lecuyer et al., 2019). (Li et al., 2019) employed Rényi divergence to obtain a higher upper bound of robust regions. Subsequently, the Neyman Pearson Lemma (NPL) was applied to find a set of worst-case points that minimizes the probability of predicting the correct classes after the data is perturbed(Cohen et al., 2019). NPL induces a tight $\ell_2$ norm robustness bound granted by smoothing the model input with Gaussian noise. To improve the certified accuracy as well as the robustness radius, a line of work focuses on training better base classifiers through deliberately designed loss functions and regularization techniques (Salman et al., 2019; Jeong & Shin, 2020; Zhai et al., 2020; Jeong et al., 2021). On the other hand, a series of methods improve the certification through specially designed smoothing noise (Súkeník et al., 2022; Cullen et al.), or even go beyond the NPL certification via techniques such as functional optimization (Dvijotham et al., 2020; Zhang et al., 2020). As its main drawback, RS is known for suffering from the curse of dimensionality when certifying $\ell_p$ attacks with $p > 2$ (Kumar et al., 2020; Yang et al., 2020). The limitation has attracted some attempts that aim at solving it (Li et al., 2022). To

increase the utility of RS for pretrained classifiers not augmented in their training phases, denoising models or model ensembles have been employed to minimize the impact of the noise on the prediction of the smoothed classifiers (Salman et al., 2020; Horváth et al., 2022). Last but not least, there are previous/concurrent works addressing the problem of certification against semantic perturbations (Fischer et al., 2020; Li et al., 2021; Hao et al., 2022).

**Randomized smoothing.** Given a $K$-class hard classifier $f(\cdot) : x \in \mathbb{R}^d \to \mathcal{Y}, x \in \mathbb{R}^d, \mathcal{Y} \in 1, ..., K$ on input $x$, a $(\sigma, f)$-classifier, which is a smoothed version of $f(\cdot)$, is defined as follows:

$$g_{\sigma,f}(x) = \arg\max_{k \in \mathcal{Y}} \Pr_{\epsilon \sim p(\epsilon|\sigma)} [f(x + \epsilon) = k], \tag{1}$$

where $\epsilon$ is a random noise sampled from a $\sigma$-parameterized probability distribution $p(\epsilon|\sigma)$. The induced robust region $R_r(x) = \{x' \in \mathbb{R}^d : f(x') = f(x), \forall \|x' - x\|_p \leq r, \}$ centered at $x$ with a radius of $r$ is often an $\ell_p$ norm ball whose radius is a function of $\sigma$ and $f$. In fact, given an adversarial space defined as a set $\mathcal{V}$ of possible perturbations $v \in \mathcal{V}$, the optimal smoothing noise distribution is found to be related to the zonotope of the vertices of $\mathcal{V}$ when $\mathcal{V}$ is a highly symmetric polytope (*e.g.,* one of $\ell_p$ norm balls) (Yang et al., 2020). $p(\epsilon|\sigma)$ can be a zero-mean Gaussain distribution with a variance of $\sigma^2$, which induces a tight $\ell_2$ norm radius around each input $x$ as follows.

$$r = \frac{\sigma}{2}[\Phi^{-1}(\underline{\Pr}[f(x + \epsilon) = y]) - \Phi^{-1}(\overline{\Pr}[f(x + \epsilon) = i])], \ \epsilon \sim p(\epsilon|\sigma), \tag{2}$$

where $\Phi^{-1}$ is the Gaussian quantile function; $\underline{\Pr}(\cdot)$ and $\overline{\Pr}(\cdot)$ are the lower bound and upper bound of their corresponding probability events; $y$ is the dominantly predicted class and $i : i \neq y$ is the runner up class. To improve $g_{\sigma,f}(x)$, a line of work trains $f$ to maximize the radius without heavily reducing the utility (Salman et al., 2019; Zhai et al., 2020; Jeong et al., 2021).

**Unrestricted perturbation.** Perturbations can be made into natural patterns free from $\ell_p$ norm constraints. These unrestricted perturbations are generally large when measured in $\ell_p$ norms but remain stealthy. We employ two distinct unrestricted attacks in the evaluation, namely, colorization-based attacks (*CAdv*) (Bhattad et al., 2019) and shadow attacks (*Shadow*) (Ghiasi et al., 2020). *CAdv* introduces a colorization model $\mathcal{C}$ with parameters $\theta$ to generate adversarial colorization schemes. In the CIELAB color space, given the light channel $x_l$ of an image $x$, user selected color hints $\mathcal{H}$ in the ab channel, and a mask $\mathcal{M}$ of the hint positions, $\mathcal{C}$ generates the pixel values $\hat{x}_{ab}$ of ab channels. By setting an adversarial loss function $L_{adv}$ (*e.g.,* adversarial PGD loss (Madry et al., 2018)) and an adversarial target $t$, the attacker can optimize $\mathcal{H}$ and $\mathcal{M}$ to result in $\mathcal{C}$ generating adversarial examples. That is

$$\mathcal{H}^*, \mathcal{M}^* = \arg\min_{\mathcal{H}, \mathcal{M}} L_{adv}(\mathcal{C}(x_l, \mathcal{H}, \mathcal{M};, \theta), t). \tag{3}$$

*CAdv* clusters an image into subareas by the pixel values in the CIELAB space and then applies a perturbation to the subarea with the highest mean entropy. In contrast, *Shadow* crafts stealthy perturbations by minimizing the total variation, mean value, and color balance of the perturbations. Furthermore, *Shadow* optimizes an expected adversarial loss over a set of samples drawn from a Gaussian distribution centered at the attacked data sample to mimic the input of a smoothed classifier. Consequently, *Shadow* examples can spoof a false robustness certificate against the smoothed classifier.

## 3 Problem definition and motivation

Increasing the robustness radius of an $\ell_p$ robustness certificate against an unrestricted adversary is hard. The space of unrestricted perturbations can be in asymmetric shapes. Henceforth, certifying an $\ell_p$ robust region against perturbations from such space requires the robustness radius to be greater than the largest possible perturbation sampled from this space. However, certifying such a large $\ell_p$ radius may bring down the utility of current certification methods due to the trade-off between the certified accuracy and the radius (Cohen et al., 2019; Yang et al., 2020). On the other hand, aligning the robustness region with the unrestricted perturbation space requires measuring and quantifying the unrestricted perturbation space, which is challenging.

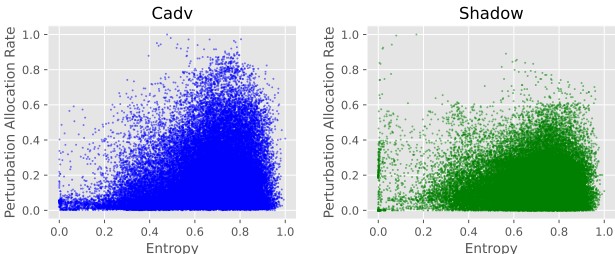

Figure 2: The relationship between the pixel-wise colorization entropy and the pixel-wise perturbation allocation rate (both values are normalized to $[0, 1]$) of adversarial examples perturbed from CIFAR10 samples. In both attacks, it can be seen that the pixels with larger entropy tend to bear more perturbations.

To address this problem, we empirically estimate the possible allocation of perturbations in different pixels of an image by certain *a prior* knowledge of the pixels. Specifically, we employ the pixel-wise entropy computed by a pretrained colorization model (Zhang et al., 2017) in our estimation. The intuition behind the strategy is that unrestricted attacks tend to conceal larger perturbations in pixels with a higher uncertainty and the uncertainty of pixel values can be indicated by the pixel-wise entropy. To support this intuition, we visualize the relationship between the entropy and perturbation allocation rate (*i.e.,*, normalized absolution perturbation values) of perturbed CIFAR10 examples by two different unrestricted attacks (*i.e.,*, Cadv and Shadow) in Figure 2. Herein, each plot contains the entropy and the perturbations of pixels from 100 randomly selected CIFAR10 images. It can be found that the perturbations are less likely to be allocated to low-entropy pixels. As the results indicate, the phenomenon can be generally observed from perturbations crafted by different attacks. Moreover, the colorization model used to compute the entropy is pretrained on ImageNet, which implies the model is generalizable between different datasets.

## 4 Concert

As discussed above, there is a trade-off between certified accuracy and robust radius. Smoothing noise with a lower variance results in a smaller robust radius but higher certified accuracy. In contrast, a higher variance may diminish the certified accuracy. Under the framework of RS, we introduce a smoothing technique used to obtain an asymmetric robustness certificate which is tighter against a non-$\ell_p$ adversarial space $\mathcal{V}$. CONCERT extends the robustness guarantee to unrestricted attacks large in $\ell_p$ norms while preserving the utility of classifiers. The workflow of CONCERT is illustrated in Figure 3.

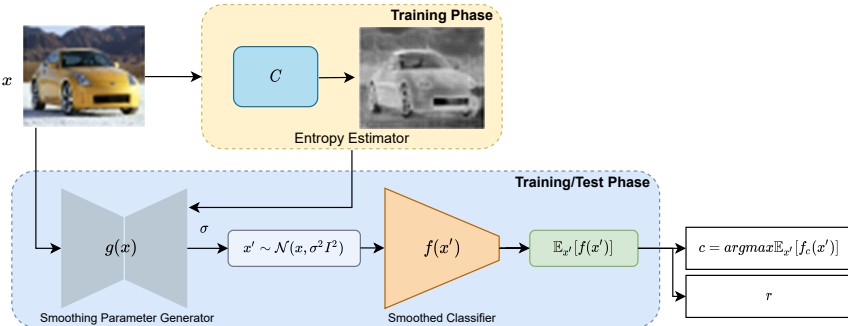

Figure 3: The workflow of CONCERT. A colorization model (*i.e.,* entropy estimator) measuring the entropy of each pixel tells the ambiguity of pixel colors. Due to the observation that adversaries tend to add more perturbations to high-entropy regions since large perturbations appear to be more natural to human auditors in these regions. A multivariate Gaussian parameterized by a generator in proportion to the entropy distribution will be optimized to maximize the certified accuracy and radius of a base classifier.

### 4.1 Worst-case classifier under sample-specific noise

When smoothing using sample-specific noise, the problem of robustness certification is commonly defined as a functional optimization as follows:

**Definition 1** (**Worst-case classifier**).

$$f^* = \arg\min_{f \in \mathcal{F}} \ \Pr[f(z) = y | z \sim p_{x+\delta}(z, \sigma_1)] \tag{4}$$
$$s.t. \ \Pr[f(z) = y | z \sim p_x(z, \sigma_0)] = T_1,$$

where $p_x(z, \sigma_1)$ and $p_{x+\delta}(z, \sigma_2)$ are likelihoods centered at $x$ and $x + \delta$, respectively. $T_1$ is the probability of predicting the top-1 (true) label, and $f^*$ as follows is the worst-case classifier in the space $\mathcal{F}$ of all possible functions.

$$f^*(z) := \mathbb{1}\left[\frac{p_{x+\delta}(z, \sigma_1)}{p_x(z, \sigma_0)} \leq t\right]. \tag{5}$$

By showing that $\Pr[f(z) = y | z \sim p_{x+\delta}(z, \sigma_1)] > 0.5, \forall \|\delta\|_p \leq R(x, p)$, any $f \in \mathcal{F}$ should be robust with respect to any $\delta$ in the $\ell_p$-norm ball with the radius $R(x, p)$. For simplicity, we note $p_x(z, \sigma_0)$ by $p_x$, $p_{x+\delta}(z, \sigma_1)$ by $p_{x+\delta}$, and $R(x, p)$ by $R$ in the following paper. The optimization problem can be fully solved by Neyman-Pearson (NP-sufficient). The details of the proof are in Appendix A. Having this definition, we will next examine the problem of robustness certification using sample-specific anisotropic noise.

### 4.2 Input-dependent robust region

We first show that the robustness certificate can be scaled dimension-wise to obtain an asymmetric robust region. Based on Definition 1, we can compute the decision boundary of the $f^*(z)$ smoothed by sample-specific Gaussian noise as follows:

**Theorem 2** (*Decision boundary of the worst-case sample-specific classifier*). *When $p_x := \mathcal{N}(x, \Sigma_0)$ and $p_{x+\delta} := \mathcal{N}(x + \delta, \Sigma_1)$, let $\Sigma_0 := [\sigma_{01}, \sigma_{02}, ..., \sigma_{0d}]^T I$ and $\Sigma_1 := [\sigma_{11}, \sigma_{12}, ..., \sigma_{1d}]^T I$ be the diagonal covariance matrices. Define $\Lambda = \sum_{i=1}^d \frac{1}{\sigma_{0i}^2 \beta_i} \delta_i^2 + 2\log(t) + 2\sum_{i=1}^d \log(\frac{\sigma_{1i}}{\sigma_{0i}})$. When $\Lambda > 0$, the decision region $D_1 := \{z : f^*(z) = 1\}$ of the worst-case classifier $f^*(\cdot)$ is an d-dimensional ellipsoid ball with foci $\{x_i - \sigma_{0i}^2 \delta_i / \beta_i\}_{i=1}^d$ and semi-axes of lengths $\{\sqrt{\alpha_i \Lambda / \beta_i}\}_{i=1}^d$. When $\Lambda < 0$, $D_1$ is the complement of the ellipsoid ball.*

Theorem 2 gives the decision region $D_1$ of the worst-case classifier. Therefore, certifying the robustness region of the randomized version of such a classifier can be viewed as computing the maximal perturbation that keeps the probability measure of the smoothing distribution on $D_1$ larger than 0.5. Let $P_x(D_1)$ and $P_{x+\delta}(D_1)$ be the the probability measures of the decision region $D_1$ under the anisotropic Gaussian centered at $x$ and $x + \delta$, respectively. Computing the probability measure within an ellipsoid ball (or its complement) is challenging. To overcome the challenge of computing $P_{x+\delta}(D_1)$, we compute their lower bounds $\underline{P_x(D_1)}$ and $\underline{P_{x+\delta}(D_1)}$ by taking the lower bound $\underline{D_1}$ of $D_1$. We define that $\underline{P_x(D_1)} := P(\underline{D_1})$ in this case. We have the following lemma for $\underline{P(D_1)}$.

**Lemma 3** (*Lower bound of anisotropic probability measure*). *Given $\Sigma_0$, $\Sigma_1$, and $D_1$ from Theorem 2, let $\underline{D_1} := \{z : \underline{f}^*(z) = 1\}$ be the decision region of the worst-case classifier $\underline{f}^* := \mathbb{1}[\frac{\underline{p_{x+\delta}}}{\underline{p_x}} \leq t]$ , where $\underline{p_x} := \mathcal{N}(x, \sigma_0^2 I)$, $\underline{p_{x+\delta}} := \mathcal{N}(x + \delta, \sigma_1^2 I)$, $\sigma_0 = \min \Sigma_0$, and $\sigma_1 = \min \Sigma_1$. The probability measure $\underline{P_{x+\delta}}(\underline{D_1})$ is a lower bound of $\underline{P_{x+\delta}}(D_1)$.*

Therefore, we convert the problem of certifying an input $x$ with sample-specific anisotropic Gaussian to a sample-specific isotropic certification problem. Please refer to Appendix A for the proof of Theorem 2 and Lemma 3. Let us define two functions $\xi_<(\delta, \sigma_1)$ and $\xi_>(\delta, \sigma_1)$ as mentioned in Appendix B. Next, the certification process is simply checking whether $\xi_<(\delta, \sigma_1) > 0.5$ or $\xi_>(\delta, \sigma_1) > 0.5$ given a $R$ with the largest possible value. We apply grid search to locate the largest value of $R$. The detailed algorithm for prediction and certification is presented in Appendix C.

### 4.3 Entropy-based robustness regularization

Theorem 2 states that the worst-case decision region can be scaled in different dimensions by adopting an anisotropic Gaussian in the smoothing process. By carefully choosing $\sigma_i$ values for different dimensions per input, the aim of CONCERT is to improve the certified accuracy while increasing the tightness of the robust region against unrestricted perturbations. Given an input $x$, we first estimate the likelihood of the adversarial perturbation in each dimension by measuring the ambiguity of the pixel values. We adopt the pretrained colorization model in Zhang et al. (2017) for the entropy estimation since their model computes a distribution of color values for each pixel in the image. The intuition of our proposal is that, *unrestricted adversaries tend to perturb high-entropy (sharp and noisy) areas rather than low-entropy (consistent and invariant) regions since such perturbations are not easily detectable.*

Let $\mathcal{C} : x \in \mathbb{R}^d \to \hat{x} \in \mathbb{R}^d, p \in \mathbb{R}^{d \times n_c}$ be the colorization model, where $\hat{x}$ is the colorized image and $p$ is a $d \times n_c$ probability matrix which contains $d$ probability vectors of length $n_c$. Each probability vector represents the distribution of color on one pixel. The entropy of the $i$-th pixel can be calculated over the distribution of $n_c$ colors as

$$\Omega_i = -\sum_{j=1}^{n_c} p_i(c_j) \log p_i(c_j), \tag{6}$$

where $p_i(c_j)$ is the probability of rendering the $i$-th pixel into the $j$-th color $c_j$. Given an $d$-dimensional image, The entropy vector $\Omega = [\Omega_1, ..., \Omega_d]^T$ is normalized to a value between $[0, 1]$ as $\Omega = (\Omega - \min \Omega)/(\max \Omega - \min \Omega)$. Altering pixels with low entropy values is not favored by adversaries since the uncertainty of colors in these regions are unambiguous to human observers, which makes the change suspicious. For example, colorization based attacks tend to perturb high-entropy regions while keeping low-entropy regions fixed (Bhattad et al., 2019). $\Omega$ is then employed to regularize the smoothing noise variance to cope with semantic adversarial spaces.

### 4.4 Optimal smoothed classifier search

Recall the definition of a smoothed $(\sigma, f)$-classifier in Equation 1. The problem of searching for the optimal $(\sigma, f)$-classifier can be reformulated to optimizing $\sigma$. In the case of anisotropic Gaussian noise, as described in Theorem 2, $\sigma$ is a vector of standard deviations. A properly smoothed classifier should maximize the certified accuracy under different robustness radii while increasing the average robustness radius.

Making $\sigma$ proportional to $\Omega$ could be an instant solution for designing the smoothing noise. However, this simple fitting strategy ignores information about the decision boundary of $f$. Therefore, it may lead to a poor-quality base classifier. Furthermore, searching for such a smoothed classifier for individual inputs can be costly. For example, previous attempts optimize $\sigma$ in a per-sample manner, which is indeed computationally demanding. Instead, we use *amortization* and model $\sigma$ with an autoencoder $G : x \in \mathbb{R}^d \to \sigma \in \mathbb{R}^d$ jointly trained with a base classifier $f$. In other words, the smoothed classifier search for an input $x$ that is its optimal $(G(x), f)$-classifier. Assuming the training data distribution $P_{data}$ such that $(x, y) \sim P_{data}$, the task here is to jointly find $f$ and $G$ that operate well in the smoothed $g_{G(x),f}$ under different $x$. In essence, $f$ and $G$ strive to maximize the following objective:

$$\max_{G,f} \underbrace{\mathbb{E}_{(x,y) \sim P_{data}} \mathbb{1}[g_{G(x),f}(x) = y]}_{\text{Classification}} + \underbrace{\mathbb{E}_{(x,y) \sim P_{data}}[r(x) \cdot \mathbb{1}[g_{G(x),f}(x) = y)]}_{\text{Robustness}}, \tag{7}$$

where $\mathbb{1}[\cdot]$ is a 0-1 loss (Zhang et al., 2019) and $r(x)$ is the robust radius of $x$. Due to the sample-specific robustness region obtained in our certification method, it is hard to find a uniform $r(x)$ to maximize since $\sigma$ is multivariate. Thus, we define the normalized radius as

$$\hat{r}(x) = \frac{1}{2}[\Phi^{-1}(\mathbb{E}_{\hat{\epsilon}} f_y(x + \sigma \hat{\epsilon})) - \Phi^{-1}(\max_{i:i \neq y} \mathbb{E}_{\hat{\epsilon}} f_i(x + \sigma \hat{\epsilon}))], \tag{8}$$

which indicates how efficient the noise is in terms of inducing a robustness guarantee. Given this definition of $\hat{r}(x)$, it is natural to optimize the volume of the robustness region. As shown by (Eiras et al., 2022), the

volume of such an ellipsoid robustness region can be obtained as

$$v(x) = \hat{r}(x)(\prod_{i=1}^{d} \sigma_i)^{\frac{1}{d}}. \tag{9}$$

Furthermore, it is well known that directly maximizing the 0-1 loss is hard so we replace it with a surrogate loss. We employ the log loss of the soft smoothed $(G(x), f)$-classifier's prediction score on the label class $y$, as our classification loss $L_c(x|f, G) = \mathbb{E}_{(x,y)\sim P_{data}} - \log \mathbb{E}_{\hat{\epsilon}} f_y[x + G(x)\hat{\epsilon}]$. However, the derivatives $\bigtriangledown_f L_c(x|f, G)$ and $\bigtriangledown_G L_c(x|f, G)$ have large variances, since

$$\bigtriangledown_f L_c(x|f, G) = \mathbb{E}_{(x,y)\sim P_{data}} - \frac{\mathbb{E}_{\hat{\epsilon}} \bigtriangledown_f f_y[x + G(x)\hat{\epsilon}]}{\mathbb{E}_{\hat{\epsilon}} f_y[x + G(x)\hat{\epsilon}]} \tag{10}$$

$$= -\mathbb{E}_{(x,y)\sim P_{data}} \mathbb{E}_{\hat{\epsilon}} [\frac{f_y[x + G(x)\hat{\epsilon}]}{\mathbb{E}_{\hat{\epsilon}} f_y[x + G(x)\hat{\epsilon}]} \bigtriangledown_f \log f_y[x + G(x)\hat{\epsilon}]],$$

where $\frac{f_y[x+G(x)\hat{\epsilon}]}{\mathbb{E}_{\hat{\epsilon}} f_y[x+G(x)\hat{\epsilon}]}$ may potentially require a large number of samples to obtain a low-variance estimate. To circumvent this problem, we can instead move the logarithm inside the expectation over $\hat{\epsilon}$ to acquire the final classification loss.

Moreover, since it is hard to capture the actual perturbation space of an unrestricted adversary, we incorporate knowledge of pixel distributions to estimate the adversarial risk of each dimension towards unrestricted perturbations. Notice that the colorization model provides the entropy of the pixels. Generally, on the one hand, pixels with higher entropy have higher risks of being altered. On the other hand, pixels with low entropy can be smoothed by noise sampled with smaller variances to account for the accuracy trade-off. Therefore, we apply a $\ell_2$-norm term of the colorization entropy $\Omega$ to penalize low-entropy dimensions of $\sigma$. Thus, the overall loss function becomes

$$L = \underbrace{\mathbb{E}_{(x,y)\sim P_{data}} - \mathbb{E}_{\hat{\epsilon}} \log f_y[x + G(x)\hat{\epsilon}]}_{\text{Classification Loss}} + \alpha \underbrace{\mathbb{E}_{(x,y)\sim P_{data}} \max\{\kappa - v(x), 0\}}_{\text{Robustness Loss}} \tag{11}$$

$$+ \beta \underbrace{\mathbb{E}_{(x,y)\sim P_{data}} \|(1 - \Omega) \odot G(x)\|_2}_{\text{Penalty}},$$

where $\alpha$, $\beta$ and $\kappa$ are hyper-parameters.

## 4.5 Training of Concert

Though there are no constraints on the generator $G$, bounding the output values of $G$ can stabilize the training process of CONCERT by avoiding extremely small/large Gaussian noise. More importantly, it bounds the noise variance at every possible input to cope with the certification process. Specifically, we apply a gated sigmoid function on the penultimate layer output $z(x)$ in $G$, given input $x$. The $i$-th output $G_i(x)$ is as follows.

$$G_i(x) = \frac{ae^{-z_i(x)} + \omega b}{1 + e^{-z_i(x)}}, \quad a, b \in \mathbb{R} \tag{12}$$

where $a, b : a \leq b$ are two constants mentioned in Section 4.2. A selected $\omega \in [a/b, 1]$ controls the maximal value of $G_i(x)$. In this way, we can ensure that $G(x) \in [a, \omega b]^d$. Next, the certification process in Section 4.2 will select $a$ and $\omega b$ as the minimal and maximal value of $\sigma_1$, respectively. This box constraint ensures the value of $\sigma$ is bounded naturally when the volume $v(x)$ is maximized while minimizing the robustness loss. Moreover, the robustness loss tends to take effect later than the classification loss since more samples are correctly classified in later stages of training. Therefore, CONCERT reduces its training overhead by using a lazy-start scheme on the robustness loss and the regularization term. That is, $\alpha$ and $\beta$ are set to zeros in early epochs. We summarize the training algorithm of CONCERT in Algorithm 1.

---

**Algorithm 1:** CONCERT Training

---

**Input**: Training distribution $P_{data}$, colorization model $C$, generator $G$, base classifier $f$, learning rate $\eta$, threshold $T$, constants $a$ and $b$, hyper-parameters $\kappa$, $\alpha$, $\beta$, and $m$

**for** $epoch \in Epochs$ **do**
    **for** $iter \in Iterations$ **do**
        Sample training batch $(x_1, y_1), ..., (x_n, y_n) \sim P_{data}$
        **for** $(x_i, y_i) \in Batch$ **do**
            $\sigma \leftarrow G(x_i)$
            Draw $m$ i.i.d. samples: $x_{i1}, ..., x_{im} \sim \mathcal{N}(x_i, \sigma^2 I)$
            Compute prediction form soft-smoothed classifier: $g(x_i) \leftarrow \frac{1}{m} \sum_{j=1}^m f(x_{ij})$
        Compute expected classification loss: $L_c \leftarrow -\frac{1}{n} \sum_{i=1}^n \log g_{y_i}(x_i)$
        **if** $epoch \geq T$ **then**
            **for** $(x_i, y_i) \in Batch$ **do**
                Compute color distribution vector: $p_i \leftarrow C(x_i)$
                Compute entropy: $\Omega_i \leftarrow -\sum_{c=1}^{n_c} p_i^c \log p_i^c$
            Compute expected robustness loss: $L_r \leftarrow \frac{1}{n} \sum_{i=1}^n \max\{\kappa - v(x_i), 0\}$
            Compute expected penalty loss: $L_p \leftarrow \frac{1}{n} \sum_{i=1}^n \|(1 - \Omega) \odot G(x_i)\|_2$
            $L \leftarrow L_c + L_r$
            Update $G$ parameters: $G \leftarrow G - \eta \frac{\partial (L + L_p)}{\partial G}$
        **else**
            $L \leftarrow L_c$
        Update $f$ parameters: $f \leftarrow f - \eta \frac{\partial L}{\partial f}$

**Output**: $G$, $f$.

---

## 5 Experiments

**Models and data.** We evaluate CONCERT on MNIST, CIFAR10, and ImageNet. We employ LeNet, ResNet-110, and ResNet-50 as the architectures of the base classifiers, respectively, for the above datasets. We include the entire training set of each dataset to train the models. The models for MNIST are tested on the entire MNIST testing set. Regarding CIFAR10, 500 samples are randomly drawn from the testing set for evaluation. The same setting applies to ImageNet. The colorization model used for calculating pixel entropy is pretrained on the ImageNet dataset. More details of the colorization model can be found in the original paper of (Zhang et al., 2017). The architecture of $G$ is presented in Appendix H.

**Metrics.** We compare our method with Cohen (Cohen et al., 2019), SmoothAdv (Salman et al., 2019), and Macer (Zhai et al., 2020). To make comparisons between isotropic and anisotropic robustness regions, we follow the definition of proxy robustness volume in previous work (Eiras et al., 2022). Note that we do not compare CONCERT with existing sample-specific certification methods (Wang et al., 2020; Alfarra et al., 2022; Chen et al., 2021; Eiras et al., 2022) since they may derive mathematically invalid certifications (Súkenık et al., 2022). Given $\hat{r}(x)$ defined in Equation 8, the proxy radius $r(x) = \min_i \sigma_i \hat{r}(x)$ is the minimal radius which is acquired on the dimension smoothed using the smallest noise variance. In other words, it is the radius of the maximal enclosed $l_2$ ball of the robustness ellipsoid. The anisotropic robustness region has a greater robustness volume to that of an isotropic region of radius $r'(x)$, if $r(x) \geq r'(x)$. The performance metrics used in the evaluation are *certified accuracy* and *robustness radius.* Since CONCERT searches for the best smoothed classifier for each individual sample, the certified accuracy here is averaged from all $(G(x), f)$-classifiers for $x$ in the test set. Certified accuracy measures how well can the $(G(x), f)$-classifiers predict and certify inputs. It is approximated from a set of testing samples. Furthermore, CONCERT induces an asymmetric robust region whose radius is difficult to be compared with the previous works that use $\ell_2$ radius. We instead measure the minimal $l_2$ radius $r(x) = \underline{\sigma} \hat{r}(x)$ in which $\underline{\sigma} = \min_{i \in \{1, ..., d\}} G(x)_i$ is the minimal generated standard deviation among $d$ dimensions. An $\ell_2$ norm ball centered at $x$ with a radius of $r(x)$ can be viewed as the largest enclosed $\ell_2$ norm ball of the actual robustness region. We measure the robustness achieved by different methods as the average minimal certified radius (AMCR) of the test set. The AMCR of the isotropic region is identical to the definition of average certified radius (ACR) used in the previous methods.

**Training details.** CONCERT is trained for 300 epochs on MNIST and CIFAR10 with an initial learning rate of 0.01 and a batch size of 128. The learning rate decays by 0.1 at the 100-th and the 200-th epochs. The lazy-start epoch $T$ is set to 200. For ImageNet, CONCERT is trained for 120 epochs with a learning rate that decays by 0.1 at the 30-th, the 60-th, and the 90-th epochs. The lazy-start epoch $T$ is set to 60. During training, we set $\alpha = 6$, $\beta = 1$, and $\kappa = 8$. The values of $(a, b)$ are selected from $\{(0.25, 0.5), (0.5, 1.0), (1.0, 1.5)\}$.

We adopt $\sigma = \{0.25, 0.5, 1\}$ in the training of the competitors. We employ two Nvidia P100 GPUs for training the models.

## 5.1 Evaluation

**Accuracy.** We first compare the certified accuracy induced by CONCERT with that of other methods, on the test datasets of MNIST and CIFAR10. We also measured the certified accuracy using 500 ImageNet samples (please refer to Appendix E for results on MNIST and ImageNet). The radii of CONCERT are the minimal $\ell_2$ radii. Therefore, the AMCR of CONCERT is actually the **worst-case** AMCR. The CIFAR10 dataset results are given in Table 1. The worst-case CONCERT achieves comparable accuracy-radius trade-offs with the benchmark schemes. When constraining the generator output in $[0.25, 0.5]$, CONCERT obtained a higher percentage of correctly classified samples with a radius between 0.25 and 0.75. When using a higher variance, *i.e.* $\underline{\sigma} = 1$, CONCERT outperforms the competitors on samples whose radius is between 0.25 and 0.5.

Table 1: Certified Accuracy on CIFAR10 dataset

| $\underline{\sigma}$ | Method | 0.00 | 0.25 | 0.50 | 0.75 | 1.00 | 1.25 | 1.50 | 1.75 | 2.00 | AMCR |
|---|---|---|---|---|---|---|---|---|---|---|---|
| 0.25 | Cohen | 0.74 | 0.59 | 0.43 | 0.25 | 0 | 0 | 0 | 0 | 0 | 0.415 |
| 0.25 | SmoothAdv | 0.72 | 0.67 | 0.58 | 0.45 | 0 | 0 | 0 | 0 | 0 | 0.536 |
| 0.25 | Macer | 0.80 | 0.70 | 0.59 | 0.43 | 0 | 0 | 0 | 0 | 0 | 0.555 |
| 0.25 | CONCERT | 0.75 | 0.69 | **0.59** | **0.43** | 0 | 0 | 0 | 0 | 0 | 0.542 |
| 0.5 | Cohen | 0.64 | 0.53 | 0.41 | 0.30 | 0.22 | 0.15 | 0.08 | 0.04 | 0 | 0.492 |
| 0.5 | SmoothAdv | 0.50 | 0.48 | 0.44 | 0.39 | 0.37 | 0.33 | 0.30 | 0.23 | 0 | 0.702 |
| 0.5 | Macer | 0.65 | 0.60 | 0.53 | 0.45 | 0.39 | 0.30 | 0.18 | 0.12 | 0 | 0.718 |
| 0.5 | CONCERT | 0.55 | 0.46 | 0.45 | 0.39 | 0.38 | 0.32 | 0.28 | 0.20 | 0 | 0.704 |
| 1.0 | Cohen | 0.47 | 0.39 | 0.34 | 0.28 | 0.21 | 0.17 | 0.14 | 0.08 | 0.05 | 0.443 |
| 1.0 | SmoothAdv | 0.45 | 0.41 | 0.38 | 0.34 | 0.32 | 0.28 | 0.25 | 0.21 | 0.19 | 0.787 |
| 1.0 | Macer | 0.45 | 0.41 | 0.37 | 0.33 | 0.32 | 0.29 | 0.25 | 0.22 | 0.18 | 0.790 |
| 1.0 | CONCERT | 0.46 | 0.40 | **0.39** | 0.31 | 0.29 | 0.26 | 0.23 | 0.20 | 0.16 | 0.781 |

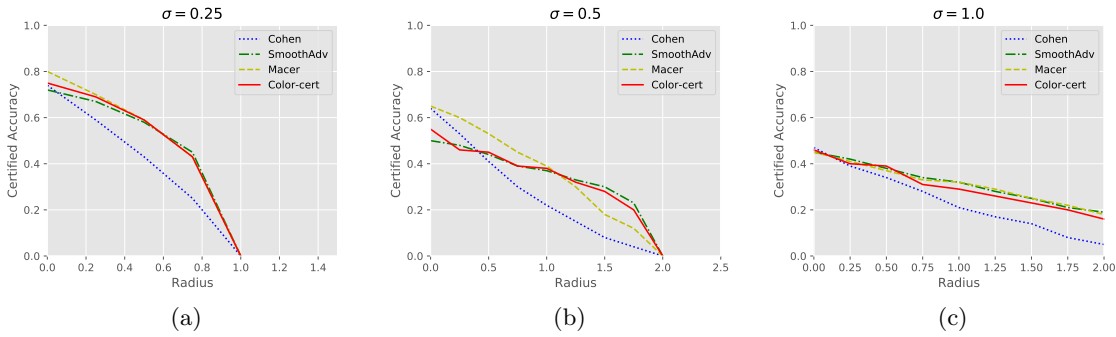

Figure 4: Certified accuracy and minimal robustness radius of different methods.

**Anisotropic robust region against unrestricted perturbations.** While maintaining an above-the-bar certified accuracy on clean samples, CONCERT distributes more robustness budget to vulnerable dimensions to mitigate unrestricted perturbations which have a large magnitude in these dimensions and exceed isotropic robustness regions. In the ideal case, the certified robust region aligns with the space of unrestricted perturbations, which means that any perturbation drawn from this space falls in the robust region and cannot change the prediction of the smoothed classifier. We call this alignment the *tightness* of the robust region towards the unrestricted perturbation space. Theoretically measuring the tightness of the robust region against the unrestricted adversary is difficult. Instead, the tightness can be empirically evaluated by the accuracy of classifying unrestricted adversarial examples sampled from the adversarial space. In this section, we measure the certified accuracy scores on perturbed samples to demonstrate that the smoothed classifiers produced by CONCERT have tighter robust regions with respect to the assumed attacks. Higher certified accuracy on the adversarial examples indicates that the verified robust regions better overlap with the underlying adversarial space. Herein, any sample correctly classified with a non-zero radius is considered as a certifiably accurate classification.

The certified accuracy of our method is compared with that of Cohen, Macer and SmoothAdv methods. The results on perturbed CIFAR10 samples are reported in Table 2 and Table 3. For *CAdv*, we crafted adversarial images against each base classifier smoothed by $\underline{\sigma} = 1.0$. In *Shadow*, we fix the variance of attack variance at 0.5 to craft adversarial examples against smoothed classifiers with $\underline{\sigma} = 0.5$. We measure the certified accuracy on different subset $S = \{x' : \|x' - x\|_2 \le \tau\}$ under various threshold $\tau$. $\tau$ is selected from a grid of $\{2.00, 2.25, 2.50, 2.75, 3.0, 4.0, 5.0, 6.0\}$ for *CAdv* and from $\{3.0, 4.0, 5.0, 6.0\}$ for *Shadow*. Each *CAdv* subset contains 100 samples. In *Shadow* attacks, due to its computation overhead, a total of 100 samples whose perturbation $\ell_2$ norm is less than 6.0 were produced. The certified accuracy is calculated as the ratio of correctly classified samples whose radii are non-zero.

Table 2: Certified Accuracy Against Non-targeted CAdv on CIFAR10

| $\tau$ | 2.00 | 2.25 | 2.50 | 2.75 | 3.0 | 4.0 | 5.0 | 6.0 |
|---|---|---|---|---|---|---|---|---|
| Cohen-1.0 | 0.20 | 0.19 | 0.18 | 0.17 | 0.15 | 0.13 | 0.09 | 0.07 |
| SmoothAdv-1.0 | 0.34 | 0.31 | 0.27 | 0.25 | 0.20 | 0.17 | 0.11 | 0.10 |
| Macer-1.0 | 0.40 | 0.30 | 0.29 | 0.23 | 0.19 | 0.16 | 0.12 | 0.07 |
| CONCERT-1.0 | 0.39 | **0.31** | 0.27 | 0.24 | **0.23** | **0.28** | **0.22** | **0.21** |

Table 3: Certified Accuracy Against Non-targeted Shadow on CIFAR10

| $\tau$ | 3.0 | 4.0 | 5.0 | 6.0 |
|---|---|---|---|---|
| Cohen-0.5 | 0 | 0.04 | 0.02 | 0.02 |
| SmoothAdv-0.5 | 0.20 | 0.09 | 0.06 | 0.06 |
| Macer-0.5 | 0.13 | 0.14 | 0.05 | 0.03 |
| CONCERT-0.5 | **0.33** | **0.29** | **0.23** | **0.15** |

Similarly, 100 ImageNet samples correctly classified by the smoothed classifiers were made into CAdv and Shadow examples, respectively. Following the same evaluation settings as that for CIFAR10, we measure the certified accuracy to demonstrate the tightness of CONCERT on ImageNet. Herein, $\tau$ is selected from a grid of $\{10, 20, 40, 80, 160, 320\}$ for *CAdv* and from $\{10, 30, 90, 270\}$ for *Shadow*. Each *CAdv* subset contains 100 samples and a total of 100 Shadow samples with $\ell_2$ perturbation less than 320 were produced. The results are as indicated by Tables 4 and 5.

Table 4: Certified Accuracy Against Non-targeted CAdv on ImageNet

| $\tau$ | 10 | 20 | 40 | 80 | 160 | 320 |
|---|---|---|---|---|---|---|
| Cohen-1.0 | 0.26 | 0.20 | 0.11 | 0 | 0 | 0 |
| SmoothAdv-1.0 | 0.31 | 0.21 | 0.14 | 0.06 | 0 | 0 |
| Macer-1.0 | 0.29 | 0.18 | 0.12 | 0.02 | 0 | 0 |
| CONCERT-1.0 | **0.38** | **0.32** | 0.14 | **0.11** | **0.09** | 0 |

Tables 2-5 demonstrates the robustness gain obtained by each smoothing technique. On CIFAR10, CONCERT achieves the highest robustness gain against *CAdv*, especially when the $\ell_2$ norm of perturbations surpasses 3.0. On the other hand, *Shadow* reduces the certified accuracy of Cohen, SmoothAdv and Macer below 0.1 under most radii. However, CONCERT preserves the certified accuracy and keeps at best a certified accuracy of 0.33. Similar trends can be observed on ImageNet.

**Ablation study.** We study the impact of hyper-parameter selection on the performance of CONCERT by observing the certified accuracy obtained under different radii, given different training hyper-parameters. The results on the CIFAR-10 dataset are plotted in Figure 5. Notice that when increasing the value of $\kappa$ from 4 to 16, CONCERT first shows higher accuracy at larger radii. However, the accuracy drops since chasing for high robustness volume increases the variance of the classifier. The same trend is observed when changing the value of $\alpha$. At last, when $\beta = 0$, CONCERT degenerates to Macer. When $\beta$ is set to 5, CONCERT tends to produce a variance of 1 on all dimensions, which leads to an almost $\ell_2$-norm-bounded robustness region. It is noticed that $\beta \in \{0.2, 1\}$ generally yields a minor decrease in the certified accuracy and average radius, which might indicate that the robustness region is reallocated to the outside of typical $\ell_p$ balls.

**Performance against adaptive attacks.** There are no adaptive attackers existing within the certified robust region for each input. However, since we apply the generator $G$ to obtain parameters for the smoothing

Table 5: Certified Accuracy Against Non-targeted Shadow on ImageNet

| $\tau$ | 10 | 30 | 90 | 270 |
|---|---|---|---|---|
| Cohen-0.5 | 0.11 | 0.11 | 0.03 | 0 |
| SmoothAdv-0.5 | 0.15 | 0.12 | 0.09 | 0 |
| Macer-0.5 | 0.10 | 0.08 | 0.03 | 0 |
| Concert-0.5 | **0.25** | **0.18** | **0.11** | 0 |

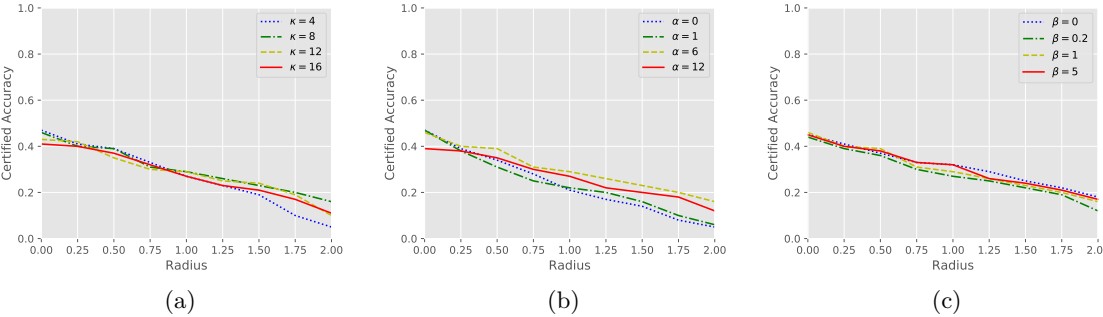

Figure 5: Certified accuracy of Concert-1.0 trained under different hyper-parameters.

noise, an adaptive attack $x + \delta$ with perturbation $\delta$ may aim at manipulating $G(x + \delta)$ to reduce the effectiveness of the defense. Notice that the smoothed classifier will be reduced to a base classifier when the variance of the smoothing noise equals zero. Therefore, a simple way of attacking is to minimize the values in $G(x')$ such that the area of each certified region shrinks. The loss function of the adaptive attacking strategy can be formulated as follows:

$$\delta^* = \arg\min_{\delta}[L_{adv} + \lambda\|G(x + \delta)\|_2], \tag{13}$$

where $L_{adv}$ is the vanilla adversarial loss of an attack (*e.g.,*, Cadv) and $\lambda$ is a constant balancing the two loss terms. To solve the optimization problem, the attacker must obtain the value of $G(x + \delta)$. When $G$ is a white box to the attacker, Equation 13 can be optimized via gradients. Otherwise, zeroth-order optimization techniques can be employed to solve the problem. Herein, we simulate the strongest adversary who has access to a white-box $G$. We test the certified accuracy of Concert-1.0 and Concert-0.5 against the adaptive versions of Cadv (Cadv-ad) and Shadow (Shadow-ad), respectively, on CIFAR10. The results are shown in Table 6. The adaptive attacks reduce the certified accuracy of Concert. However, it can be observed

Table 6: Certified Accuracy Against Adaptive Attacks on CIFAR10

| $\tau$ | 2.0 | 3.0 | 4.0 | 5.0 | 6.0 |
|---|---|---|---|---|---|
| Cadv-ad | 0.28 | 0.20 | 0.17 | 0.15 | 0.10 |
| Shadow-ad | 0.15 | 0.13 | 0.10 | 0.08 | 0.05 |

that the performance of Concert under the adaptive attacks is still better than or on par with that of the competitors under vanilla attacks. This is due to the limit placed on the output of $G(\cdot)$. The adaptive attacker can at most reduce the generated variance to a value close to $a^2$, in which case Concert may randomize classifier inputs using an isotropic noise.

## 6   Discussion

**Effectiveness of Concert.** Allocating smoothing noise in a context-aware manner is important in defending against unrestricted perturbations. Unrestricted perturbations are often blended into particular image areas, which preserves the naturalness of the perturbed images. In return, the perturbed images can tolerate substantial perturbations without compromising the stealthiness of the adversarial attack. Concert assumes a higher noise level on high-entropy pixels to enhance the robustness of the smoothed classifier in vulnerable

input dimensions. The experimental results demonstrate the versatility and efficacy of the entropy-guided smoothing strategy, as it has been successfully applied to a diverse range of attacking methods. This highlights the importance of pixel entropy as a reliable metric for determining the susceptibility of pixels to perturbation, confirming its utility as a guiding principle in smoothing techniques. This claim is further supported by a comparison of the entropy distribution and the perturbation intensity based on semantic perturbations generated against base/smoothed classifiers. It can be observed in Figure 6 that the perturbed areas are associated with high pixel entropy values in both *CAdv* and Shadow attacks. Moreover, the entropy distribution calculated on ImageNet is well generalized to CIFAR10 in each attack.

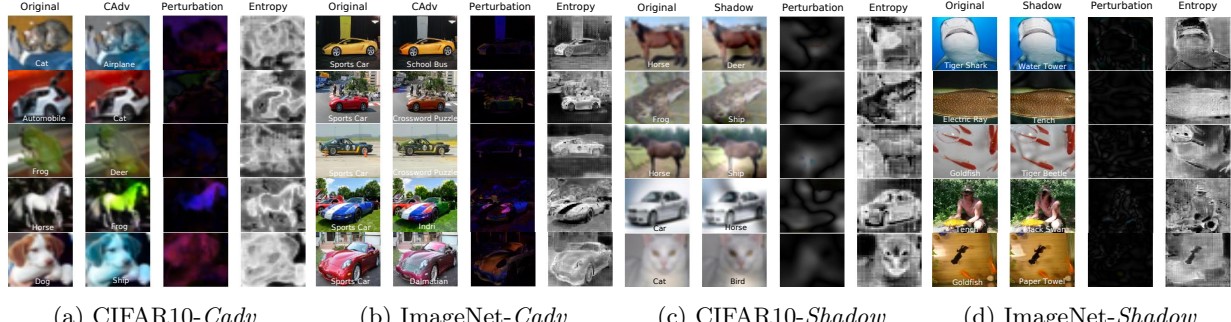

(a) CIFAR10-*Cadv*    (b) ImageNet-*Cadv*    (c) CIFAR10-*Shadow*    (d) ImageNet-*Shadow*

Figure 6: On the left are *Cadv* perturbations and the pixel-wise entropy on (a) CIFAR10 and (b) ImageNet samples. It can be seen that the perturbations tend to be located at high-entropy (i.e. ambiguous) regions for samples from both datasets, despite the entropy being estimated from ImageNet. Moreover, *Shadow* perturbations and the pixel-wise entropy on (c) CIFAR10 and (d) ImageNet samples are illustrated. These perturbations remain concentrated in high-entropy regions. Notice that *Shadow* does not involve any entropy information in the attack. The correlation between the perturbed region and the entropy distribution shows that the colorization entropy can be generalized to *Shadow* perturbations.

**Limitations.** CONCERT regularizes generated noise variance using colorization-based entropy, given the observation that semantic perturbations are more likely to occur in the color-ambiguous areas of images than in the other areas. Therefore, the correlation between the entropy and the perturbation distribution is decisive for the defensive performance. Furthermore, since CONCERT jointly trains the generator and the base classifier, additional overhead is incurred when compared to isotropic certification methods. Learning a conditional distribution between the noise variances and high-dimensional inputs could be challenging. Distilling better representations of the entropy information to assist the design of noise variance can help when scaling CONCERT to large inputs.

## 7   Conclusion

We propose CONCERT as a data-dependent anisotropic smoothing mechanism in certified robustness which allocates more robustness radius to vulnerable dimensions of input samples. By regularizing the variance of the smoothing noise using pixel-wise entropy, larger noise is applied to high-entropy areas where large perturbations are more likely to strike. CONCERT is applied to Gaussian smoothed classifiers trained on various datasets. Instead of improving a single base classifier, we search for the optimal smoothing noise for each input and train the base classifier to operate under different smoothed classifiers. CONCERT obtains on par certified accuracy and superior $\ell_2$ radius on benign inputs, compared to the state-of-the-art methods in the field. The remarkable feature of the CONCERT approach is its exceptional ability to generalize pixel-wise entropy information to various perturbations, leading to a significantly higher level of certified accuracy and a considerably tighter robustness region, even in the face of unrestricted perturbations. However, CONCERT might be computationally expensive for domain-specific models/datasets in which the entropy distribution needs to be computed prior to generating smoothing noise. To address this limitation, future research should explore more efficient and generic methods of estimating pixel ambiguity, in order to make the CONCERT approach more widely applicable.

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

## A   Proofs

**Definition 1** (**Worst-case classifier**)**.**

$$f^* = \underset{f \in \mathcal{F}}{\arg\min} \ \Pr[f(z) = y | z \sim p_{x+\delta}(z, \sigma_1)] \tag{4}$$

$$s.t. \ \Pr[f(z) = y | z \sim p_x(z, \sigma_0)] = T_1,$$

*where $p_x(z, \sigma_1)$ and $p_{x+\delta}(z, \sigma_2)$ are likelihoods centered at $x$ and $x + \delta$, respectively. $T_1$ is the probability of predicting the top-1 (true) label, and $f^*$ as follows is the worst-case classifier in the space $\mathcal{F}$ of all possible functions.*

$$f^*(z) := \mathbb{1}\big[\frac{p_{x+\delta}(z, \sigma_1)}{p_x(z, \sigma_0)} \le t\big]. \tag{5}$$

*Proof.* Define the objective $J(x, y, \delta, \sigma) = \Pr[f(z) = y | z \sim p_{x+\delta}(z, \sigma_1)]$, then

$$
\begin{aligned}
J(x, y, \delta, \sigma) &= \int_{z \in \mathbb{R}^d} \mathbb{1}[f(z) = y] \Pr[z \sim p_{x+\delta}(z, \sigma_1)] \\
&= \int_{z \in \mathbb{R}^d} \frac{\Pr[z \sim p_{x+\delta}(z, \sigma_1)]}{\Pr[z \sim p_x(z, \sigma_0)]} \mathbb{1}[f(z) = y] \Pr[z \sim p_x(z, \sigma_0)] \\
&= \int_{z \in \mathbb{R}^d} \frac{p_{x+\delta}(z, \sigma_1)}{p_x(z, \sigma_0)} \mathbb{1}[f(z) = y] p_x(z, \sigma_0) dz.
\end{aligned}
$$

Since the constraint says $\int_{z \in \mathbb{R}^d} \mathbb{1}[f(z) = y] p_x(z, \sigma_0) dz = T_1$, minimizing $J(x, y, \delta, \sigma)$ is equivalent to sorting $z \in \mathbb{R}^d$ based on the values of $\frac{p_{x+\delta}(z, \sigma_1)}{p_x(z, \sigma_0)}$, from the smallest to the largest, and pick $z$ with the smallest likelihood ratio for the integration. The worst classifier $f^*(z)$ is thus:

$$f^*(z) := \mathbb{1}\big[\frac{p_{x+\delta}(z, \sigma_1)}{p_x(z, \sigma_0)} \le t\big], \tag{14}$$

where $0 < t < +\infty$ is some threshold selected to satisfy the constraint. The minimum is $f$-independent and the problem is NP-sufficient. $\square$

**Theorem 2** (*Decision boundary of the worst-case sample-specific classifier*)**.** *When $p_x := \mathcal{N}(x, \Sigma_0)$ and $p_{x+\delta} := \mathcal{N}(x + \delta, \Sigma_1)$, let $\Sigma_0 := [\sigma_{01}, \sigma_{02}, ..., \sigma_{0d}]^T I$ and $\Sigma_1 := [\sigma_{11}, \sigma_{12}, ..., \sigma_{1d}]^T I$ be the diagonal covariance matrices. Define $\Lambda = \sum_{i=1}^{d} \frac{1}{\sigma_{0i}{}^2 \beta_i} \delta_i{}^2 + 2\log(t) + 2\sum_{i=1}^{d} \log(\frac{\sigma_{1i}}{\sigma_{0i}})$. When $\Lambda > 0$, the decision region $D_1 := \{z : f^*(z) = 1\}$ of the worst-case classifier $f^*(\cdot)$ is an $d$-dimensional ellipsoid ball with foci $\{x_i - \sigma_{0i}{}^2\delta_i/\beta_i\}_{i=1}^{d}$ and semi-axes of lengths $\{\sqrt{\alpha_i \Lambda/\beta_i}\}_{i=1}^{d}$. When $\Lambda < 0$, $D_1$ is the complement of the ellipsoid ball.*

*Proof.* When $p_x := \mathcal{N}(x, \Sigma_0)$ and $p_{x+\delta} := \mathcal{N}(x + \delta, \Sigma_1)$, let $\Sigma_0 := \sigma_0 I$ and $\Sigma_1 := \sigma_1 I$ be the diagonal covariance matrices in which $\sigma_0 = [\sigma_{01}, \sigma_{02}, ..., \sigma_{0d}]^T$ and $\sigma_1 = [\sigma_{11}, \sigma_{12}, ..., \sigma_{1d}]^T$. The decision region of the true class $D_1 := \{z : f^*(z) = 1\}$ can be further derived as $z$ following the condition below:

$$
\begin{aligned}
& \frac{1}{(2\pi)^{d/2} \prod_{i=1}^{d} \sigma_{1i}} \exp(-\frac{1}{2} \sum_{i=1}^{d} \frac{(z_i - x_i - \delta_i)^2}{\sigma_{1i}{}^2}) \le \frac{t}{(2\pi)^{d/2} \prod_{i=1}^{d} \sigma_{0i}} \exp(-\frac{1}{2} \sum_{i=1}^{d} \frac{(z_i - x_i)^2}{\sigma_{0i}{}^2}) \\
\Longleftrightarrow \ & -\frac{1}{2} \sum_{i=1}^{d} \frac{(z_i - x_i - \delta_i)^2}{\sigma_{1i}{}^2} \le \log(t) + \sum_{i=1}^{d} \log(\frac{\sigma_{1i}}{\sigma_{0i}}) - \frac{1}{2} \sum_{i=1}^{d} \frac{(z_i - x_i)^2}{\sigma_{0i}{}^2} \\
\Longleftrightarrow \ & \sum_{i=1}^{d} \big[\frac{(z_i - x_i)^2}{\sigma_{0i}{}^2} - \frac{(z_i - x_i - \delta_i)^2}{\sigma_{1i}{}^2}\big] \le 2\log(t) + 2\sum_{i=1}^{d} \log(\frac{\sigma_{1i}}{\sigma_{0i}}) \\
\Longleftrightarrow \ & \sum_{i=1}^{d} \frac{(\sigma_{1i}{}^2 - \sigma_{0i}{}^2)z_i{}^2 - 2(\sigma_{1i}{}^2 - \sigma_{0i}{}^2)x_i z_i + (\sigma_{1i}{}^2 - \sigma_{0i}{}^2)x_i{}^2 - \sigma_{0i}{}^2(\delta_i{}^2 - 2\delta_i z_i + 2x_i \delta_i)}{\sigma_{0i}{}^2 \sigma_{1i}{}^2} \\
& \le 2\log(t) + 2\sum_{i=1}^{d} \log(\frac{\sigma_{1i}}{\sigma_{0i}}).
\end{aligned}
\tag{15}
$$

Let $\alpha_i = \sigma_{0i}{}^2 \sigma_{1i}{}^2$ and $\beta_i = \sigma_{1i}{}^2 - \sigma_{0i}{}^2$. Substituting them into the above inequation yields

$$
\begin{aligned}
\iff & \sum_{i=1}^{d} \frac{z_i{}^2 - 2x_i z_i + x_i{}^2 - \frac{\sigma_{0i}{}^2}{\beta_i}(\delta_i{}^2 - 2\delta_i z_i + 2x_i \delta_i)}{\alpha_i / \beta_i} \leq 2\log(t) + 2\sum_{i=1}^{d} \log(\frac{\sigma_{1i}}{\sigma_{0i}}) \\
\iff & \sum_{i=1}^{d} \frac{[z_i - (x_i - \frac{\sigma_{0i}{}^2}{\beta_i}\delta_i)]^2}{\alpha_i / \beta_i} \leq \sum_{i=1}^{d} \frac{1}{\sigma_{0i}{}^2 \beta_i}\delta_i{}^2 + 2\log(t) + 2\sum_{i=1}^{d} \log(\frac{\sigma_{1i}}{\sigma_{0i}}).
\end{aligned}
\tag{16}
$$

The RHS is actually $\Lambda$. When $\Lambda > 0$, the above inequation becomes

$$
\sum_{i=1}^{d} \frac{[z_i - (x_i - \frac{\sigma_{0i}{}^2}{\beta_i}\delta_i)]^2}{\alpha_i \Lambda / \beta_i} \leq 1.
\tag{17}
$$

The decision region is immediately the inner space of an $d$-dimensional ellipsoid ball with foci $\{x_i - \sigma_{0i}{}^2 \delta_i / \beta_i\}_{i=1}^{d}$ and semi-axes of length $\sqrt{\alpha_i \Lambda / \beta_i}$ in the $i$-th dimension. Otherwise, when $\Lambda < 0$, the The decision region is the complement of the $d$-dimensional ellipsoid ball. $\square$

**Lemma 3** (*Lower bound of anisotropic probability measure*). *Given $\Sigma_0$, $\Sigma_1$, and $D_1$ from Theorem 2, let $\underline{D_1} := \{z : \underline{f}^*(z) = 1\}$ be the decision region of the worst-case classifier $\underline{f}^* := \mathbb{1}[\frac{\underline{p}_{x+\delta}}{\underline{p}_x} \leq t]$, where $\underline{p}_x := \mathcal{N}(x, \sigma_0{}^2 I)$, $\underline{p}_{x+\delta} := \mathcal{N}(x+\delta, \sigma_1{}^2 I)$, $\sigma_0 = \min \Sigma_0$, and $\sigma_1 = \min \Sigma_1$. The probability measure $\underline{P}_{x+\delta}(\underline{D_1})$ is a lower bound of $\underline{P}_{x+\delta}(D_1)$.*

*Proof.* The decision region of the correct class satisfies

$$
\begin{aligned}
& \frac{\underline{p}_{x+\delta}}{\underline{p}_x} \leq t \\
\iff & \frac{1}{(2\pi)^{d/2}\sigma_1{}^d} \exp[-\frac{1}{2\sigma_1{}^2}\sum_{i=1}^{d}(z_i - x_i - \delta_i)^2] \leq \frac{t}{(2\pi)^{d/2}\sigma_0{}^d} \exp[-\frac{1}{2\sigma_0{}^2}\sum_{i=1}^{d}(z_i - x_i)^2].
\end{aligned}
\tag{18}
$$

Without loss of generality, when considering this isotropic Gaussian, we can let $x = \mathbf{0}$ and the perturbation can be set to a $d$-dimensional vector $\delta = [a, 0, ..., 0]^T$. Then we have

$$
\begin{aligned}
& \frac{1}{(2\pi)^{d/2}\sigma_1{}^d} \exp[-\frac{1}{2\sigma_1{}^2}(\sum_{i=2}^{d} z_i{}^2 + (z_1 - a)^2)] \leq \frac{t}{(2\pi)^{d/2}\sigma_0{}^d} \exp[-\frac{1}{2\sigma_0{}^2}\sum_{i=1}^{d} z_i{}^2] \\
\iff & \exp[-\frac{1}{2\sigma_1{}^2}(\sum_{i=2}^{d} z_i{}^2 + (z_1 - a)^2)] \leq t(\frac{\sigma_1}{\sigma_0})^d \exp[-\frac{1}{2\sigma_0{}^2}\sum_{i=1}^{d} z_i{}^2] \\
\iff & -\frac{1}{2\sigma_1{}^2}(\sum_{i=2}^{d} z_i{}^2 + (z_1 - a)^2) \leq \log(t) + d\log(\frac{\sigma_1}{\sigma_0}) - \frac{1}{2\sigma_0{}^2}\sum_{i=1}^{d} z_i{}^2 \\
\iff & \sigma_1{}^2 \sum_{i=1}^{d} z_i{}^2 - \sigma_0{}^2[\sum_{i=2}^{d} z_i{}^2 + (z_1 - a)^2] \leq 2\sigma_0{}^2 \sigma_1{}^2[\log(t) + d\log(\frac{\sigma_1}{\sigma_0})] \\
\iff & (\sigma_1{}^2 - \sigma_0{}^2)\sum_{i=2}^{d} z_i{}^2 + (\sigma_1{}^2 - \sigma_0{}^2)z_1{}^2 + 2\sigma_0{}^2 a z_1 - \sigma_0{}^2 a^2 \leq 2\sigma_0{}^2 \sigma_1{}^2[\log(t) + d\log(\frac{\sigma_1}{\sigma_0})].
\end{aligned}
\tag{19}
$$

When $\sigma_0 < \sigma_1$, Inequation 19 becomes

$$
\begin{aligned}
& \sum_{i=2}^{d} z_i{}^2 + z_1{}^2 + \frac{2a\sigma_0{}^2}{\sigma_1{}^2 - \sigma_0{}^2}z_1 - \frac{a^2\sigma_0{}^2}{\sigma_1{}^2 - \sigma_0{}^2} \leq 2\frac{\sigma_0{}^2\sigma_1{}^2}{\sigma_1{}^2 - \sigma_0{}^2}[\log(t) + d\log(\frac{\sigma_1}{\sigma_0})] \\
\iff & \sum_{i=2}^{d} z_i{}^2 + (z_1 + \frac{a\sigma_0{}^2}{\sigma_1{}^2 - \sigma_0{}^2})^2 \leq \frac{\sigma_0{}^2\sigma_1{}^2}{(\sigma_1{}^2 - \sigma_0{}^2)^2}a^2 + 2\frac{\sigma_0{}^2\sigma_1{}^2}{\sigma_1{}^2 - \sigma_0{}^2}[\log(t) + d\log(\frac{\sigma_1}{\sigma_0})].
\end{aligned}
\tag{20}
$$

Therefore, the region $\underline{D_1}$ as above is a ball centered at $s_< = x + \frac{\sigma_1^2}{\sigma_1^2 - \sigma_0^2}\delta$ with a radius of $r_< = \sqrt{\frac{\sigma_0^2\sigma_1^2}{(\sigma_1^2-\sigma_0^2)^2}\|\delta\|^2 + 2\frac{\sigma_0^2\sigma_1^2}{\sigma_1^2-\sigma_0^2}[\log(t)+d\log(\frac{\sigma_1}{\sigma_0})]}$. Herein, $<$ and $>$ mark the relationship between $\sigma_0$ and $\sigma_1$.

Similarly, when $\sigma_0 > \sigma_1$, Inequation 19 becomes

$$\sum_{i=2}^{d} z_i^2 + z_1^2 - \frac{2a\sigma_0^2}{\sigma_0^2-\sigma_1^2}z_1 + \frac{a^2\sigma_0^2}{\sigma_0^2-\sigma_1^2} \geq 2\frac{\sigma_0^2\sigma_1^2}{\sigma_0^2-\sigma_1^2}[\log(\frac{1}{t})+d\log(\frac{\sigma_0}{\sigma_1})]$$

$$\Longleftrightarrow \sum_{i=2}^{d} z_i^2 + (z_1 - \frac{a\sigma_0^2}{\sigma_0^2-\sigma_1^2})^2 \geq \frac{\sigma_0^2\sigma_1^2}{(\sigma_0^2-\sigma_1^2)^2}a^2 + 2\frac{\sigma_0^2\sigma_1^2}{\sigma_0^2-\sigma_1^2}[\log(\frac{1}{t})+d\log(\frac{\sigma_0}{\sigma_1})]$$

(21)

Now, $\underline{D_1}$ is the complement of a ball centered at $s_> = x - \delta\frac{\sigma_1^2}{\sigma_0^2-\sigma_1^2}$ with a radius of $r_> = \sqrt{\frac{\sigma_0^2\sigma_1^2}{(\sigma_0^2-\sigma_1^2)^2}\|\delta\|^2 + 2\frac{\sigma_0^2\sigma_1^2}{\sigma_0^2-\sigma_1^2}[\log(\frac{1}{t})+d\log(\frac{\sigma_0}{\sigma_1})]}$. According to Theorem 2, the $i$-th semi-axes of region $D_1$ are

$$\sqrt{\frac{\alpha_i\Lambda}{\beta_i}}$$

$$= \sqrt{\frac{\sigma_{0i}^2\sigma_{1i}^2}{\sigma_{1i}^2-\sigma_{0i}^2}\sum_{i=1}^{d}\frac{1}{\sigma_{0i}^2(\sigma_{1i}^2-\sigma_{0i}^2)}\delta_i^2 + \frac{\sigma_{0i}^2\sigma_{1i}^2}{\sigma_{1i}^2-\sigma_{0i}^2}d\log(t) + 2\frac{\sigma_{0i}^2\sigma_{1i}^2}{\sigma_{1i}^2-\sigma_{0i}^2}\sum_{i=1}^{d}\log(\frac{\sigma_{1i}}{\sigma_{0i}})}.$$

(22)

Comparing Equation 22 with the radii $r_>$ and $r_<$, it is easy to see that $\underline{D_1}$ is the largest enclosed hyper-ball of the hyper-ellipsoid $D_1$.

When smoothing using the Gaussian $p_x$, the probability measure $P_x(\underline{D_1})$ of the region is the CDF of Noncentral Chi-square Distribution (NCD). According to the proof of (Súkeník et al., 2022), there is:

$$\underline{P}_x(\underline{D_1}) = 1 - \Phi_{\chi^2}(\frac{\sigma_0^2}{(\sigma_1^2-\sigma_0^2)^2}\|\delta\|^2, \frac{R_{<,>}^2}{\sigma_0^2}),$$

$$\underline{P}_{x+\delta}(\underline{D_1}) = 1 - \Phi_{\chi^2}(\frac{\sigma_1^2}{(\sigma_1^2-\sigma_0^2)^2}\|\delta\|^2, \frac{R_{<,>}^2}{\sigma_1^2}).$$

(23)

Herein, $\underline{P}_x(\underline{D_1}) = T_1$ ($T_1$ is the top-1 probability defined in Definition 1) is the constraint based on our observation of the classifier at $x$. There is:

$$R^2 = \Phi_{\chi^2}^{-1}(\frac{\sigma_1^2}{(\sigma_1^2-\sigma_0^2)^2}\|\delta\|^2, 1-T_1).$$

(24)

Since $\underline{D_1} \subseteq D_1$, due to the monotonicity of CDF, we have:

$$\underline{P}_{x+\delta}(\underline{D_1}) \leq \underline{P}_{x+\delta}(D_1),$$

(25)

which concludes the proof. $\qquad\square$

## B  Definintions of $\xi$ functions

Given $\sigma_0$ and $\sigma_1$ from Lemma 3, we defined $R_>$ as the radius of $\underline{D_1}$ when $\sigma_0 > \sigma_1$. $R_<$ is the radius when $\sigma_0 > \sigma_1$.

$$R_> = \sqrt{\frac{\sigma_0^2\sigma_1^2}{(\sigma_0^2-\sigma_1^2)^2}\|\delta\|^2 + 2d\frac{\sigma_0^2\sigma_1^2}{\sigma_0^2-\sigma_1^2}\log(\frac{\sigma_0}{\sigma_1}) + \frac{2\sigma_0^2\sigma_1^2}{\sigma_0^2-\sigma_1^2}\log(t)},$$

$$R_< = \sqrt{\frac{2\sigma_0^4-\sigma_0^2\sigma_1^2}{(\sigma_1^2-\sigma_0^2)^2}\|\delta\|^2 + 2d\frac{\sigma_0^2\sigma_1^2}{\sigma_1^2-\sigma_0^2}\log(\frac{\sigma_1}{\sigma_0}) - \frac{2\sigma_0^2\sigma_1^2}{\sigma_1^2-\sigma_0^2}\log(t)}.$$

(26)

Next, we can define two $\xi$ functions as follows:

$$\xi_<(\delta, \sigma_1) := \underline{P}_{x+\delta}(\underline{D}_1)$$
$$= 1 - \Phi_{\chi^2}\left(\frac{\sigma_1^2}{(\sigma_1^2 - \sigma_0^2)^2}\|\delta\|^2, \frac{R_<^2}{\sigma_1^2}\right) \tag{27}$$

and

$$\xi_>(\delta, \sigma_1) := \underline{P}_{x+\delta}(\underline{D}_1)$$
$$= 1 - \Phi_{\chi^2}\left(\frac{\sigma_1^2}{(\sigma_1^2 - \sigma_0^2)^2}\|\delta\|^2, \frac{R_>^2}{\sigma_1^2}\right), \tag{28}$$

where $\sigma_0$ and $\sigma_1$ come from Lemma 3. Similar to another work (Súkenık et al., 2022), $\xi_<(\delta, \sigma_1)$ and $\xi_>(\delta, \sigma_1)$ demonstrate the monotonicity property with respect to smoothing variance values in the range $[\sigma_0, \sigma_1]$, $if\ \sigma_0 < \sigma_1$ or $[\sigma_1, \sigma_0]$, $if\ \sigma_0 > \sigma_1$. In this sense, we set $\sigma_1 \in [a, b]$, where $a$ and $b$ are two constants bounding the worst $\sigma_1$ we would get. Note that, in lieu of setting $\sigma_1$ to be a semi-elastic function of $\sigma_0$ and $R_>/R_<$ like the previous work does (Súkenık et al., 2022), we assume $\sigma_1$ is irrelevant to $R_>/R_<$.

## C  Practical certification algorithms

We present the algorithms for prediction and certification in the following pseudocode.

---

**Algorithm 2:** CONCERT Prediction and Certification (following notations in (Cohen et al., 2019))

---

**Initialization**: Base classifier $f$, noise variance generator $G$, input data point $x$, sampling number $n$ and $n_0$, confidence level $\alpha$

**Function** $Predict(f, G, n, \alpha)$:
    $\sigma_0 \leftarrow G(x)$
    $counts \leftarrow SampleUnderNoise(f, x, n, \sigma_0)$
    $\hat{I}_A, \hat{I}_B \leftarrow Top-2classindicesincounts$
    $n_A, n_B \leftarrow counts(\hat{I}_A), counts(\hat{I}_B)$
    **if** $BinomPValue(n_A, n_A + n_B, 0.5) \leq \alpha$
        **return** $\hat{I}_A$
    **else**
        **return** $ABSTAIN$

**Function** $Certify(f, G, n_0, n, \alpha)$:
    $\sigma_0 \leftarrow G(x)$
    $counts_0 \leftarrow SampleUnderNoise(f, x, n_0, \sigma_0)$
    $\hat{I}_A \leftarrow Top-1classindicesincounts$
    $counts \leftarrow SampleUnderNoise(f, x, n, \sigma_0)$
    $\underline{p_A} \leftarrow LowerConfBound(counts[\hat{I}_A], n, 1 - \alpha)$
    **if** $\underline{p_A} > 0.5$
        **return** prediction $\hat{I}_A$ and radius $ComputerCertifiedRadius(\sigma_0, r, N, \underline{p_A}, num_steps, num_grid)$
    **else**
        **return** $ABSTAIN$

**Function** $ComputerCertifiedRadius(\sigma_0, a, b, \omega, N, \underline{p_A}, num_{grid})$:
    $radii \leftarrow (num_{grid})$
    **for** $R$ in $radii$ **do**
    $\sigma_{1<} \leftarrow a$
    $\sigma_{1>} \leftarrow \omega b$
    $check_< \leftarrow \xi_<(\delta, \sigma_{1<})$
    $check_> \leftarrow \xi_>(\delta, \sigma_{1>})$
    **if** $\min\{check_<,\ check_>\} < 0.5$
        **break**
    **return** $\delta$

---

## D  Conversion between RGB and CIELAB

CIELAB, or Lab, employs a combination of lightness $L$ with ab color channels $a$ and $b$ to represent pixel values. During converting color representation from RGB to $Lab$, CIEXYZ colors are used as a medium for the conversion, vice versa. Given the chromaticity coordinates $(x_r, y_r)$, $(x_g, y_g)$, and $(x_b, y_b)$ of an RGB

system and its reference white $(X_w, Y_w, Z_w)$, it is transformed to CIEXYZ as

$$\begin{bmatrix} X \\ Y \\ Z \end{bmatrix} = M \begin{bmatrix} R \\ G \\ B \end{bmatrix}, M = \begin{bmatrix} S_r X_r & S_g X_g & S_b X_b \\ S_r Y_r & S_g Y_g & S_b Y_b \\ S_r Z_r & S_g Z_g & S_b Z_b \end{bmatrix} \tag{29}$$

where $X_i = x_i/y_i$, $Y_i = 1$, $Z_i = (1 - x_i - y_i)/y_i, \forall\, i \in \{r, g, b\}$, and

$$\begin{bmatrix} S_r \\ S_g \\ S_b \end{bmatrix} = \begin{bmatrix} X_r & X_g & X_b \\ Y_r & Y_g & Y_b \\ Z_r & Z_g & Z_b \end{bmatrix}^{-1} \begin{bmatrix} X_w \\ Y_w \\ Z_w \end{bmatrix}. \tag{30}$$

Subsequently, given the CIEXYZ reference white $(X_{ref}, Y_{ref}, Z_{ref})$, Lab values can be calculated as

$$L = 116 f_y - 16 \tag{31}$$
$$a = 500(f_x - f_y)$$
$$b = 200(f_y - f_z).$$

Herein,

$$f_x = \begin{cases} \sqrt[3]{x_{ref}} & if\ x_{ref} > \epsilon \\ \frac{\kappa x_{ref} + 16}{116} & otherwise \end{cases}, \tag{32}$$

$$f_y = \begin{cases} \sqrt[3]{y_{ref}} & if\ y_{ref} > \epsilon \\ \frac{\kappa y_{ref} + 16}{116} & otherwise \end{cases}, \ and \tag{33}$$

$$f_z = \begin{cases} \sqrt[3]{z_{ref}} & if\ z_{ref} > \epsilon \\ \frac{\kappa z_{ref} + 16}{116} & otherwise \end{cases} \tag{34}$$

where $x_{ref} = X/X_{ref}$, $y_{ref} = Y/Y_{ref}$, and $z_{ref} = Z/Z_{ref}$. $\epsilon = 0.008856$ and $\kappa = 903.3$. Lab colors can be converted to RGB by taking the inverse of the above process. However, the Lab gamut is larger than the RGB gamut, which means that a Lab chromaticity coordinates may not find its counterpart in the RGB space.

## E More experimental results

We present certification results on MNIST and ImageNet in this section.

Table 7: Certified Accuracy on MNIST

| $\sigma$ | Method | 0.0 | 0.25 | 0.50 | 0.75 | 1.00 | 1.25 | 1.50 | 1.75 | 2.0 | AMCR |
|---|---|---|---|---|---|---|---|---|---|---|---|
| 0.25 | Cohen | 0.99 | 0.98 | 0.97 | 0.93 | 0 | 0 | 0 | 0 | 0 | 0.912 |
| 0.25 | SmoothAdv | 0.99 | 0.99 | 0.98 | 0.96 | 0 | 0 | 0 | 0 | 0 | 0.931 |
| 0.25 | Macer | 0.99 | 0.99 | 0.97 | 0.95 | 0 | 0 | 0 | 0 | 0 | 0.920 |
| 0.25 | Concert | 0.99 | 0.99 | 0.97 | 0.93 | 0 | 0 | 0 | 0 | 0 | 0.918 |
| 0.5 | Cohen | 0.99 | 0.98 | 0.97 | 0.94 | 0.90 | 0.82 | 0.67 | 0.43 | 0 | 1.551 |
| 0.5 | SmoothAdv | 0.99 | 0.98 | 0.97 | 0.95 | 0.93 | 0.88 | 0.81 | 0.68 | 0 | 1.683 |
| 0.5 | Macer | 0.99 | 0.98 | 0.96 | 0.94 | 0.90 | 0.84 | 0.73 | 0.54 | 0 | 1.595 |
| 0.5 | Concert | 0.99 | **0.99** | 0.97 | **0.96** | 0.90 | 0.83 | 0.69 | 0.51 | 0 | 1.583 |
| 1.0 | Cohen | 0.96 | 0.93 | 0.91 | 0.87 | 0.80 | 0.70 | 0.59 | 0.46 | 0.33 | 1.616 |
| 1.0 | SmoothAdv | 0.96 | 0.94 | 0.91 | 0.87 | 0.81 | 0.74 | 0.65 | 0.54 | 0.43 | 1.781 |
| 1.0 | Macer | 0.92 | 0.89 | 0.84 | 0.78 | 0.72 | 0.64 | 0.56 | 0.46 | 0.40 | 1.641 |
| 1.0 | Concert | 0.95 | 0.93 | 0.90 | 0.83 | 0.80 | 0.71 | 0.56 | 0.39 | 0.31 | 1.592 |

Table 8: Certified Accuracy on ImageNet

| $\underline{\sigma}$ | Method | 0.0 | 0.5 | 1.0 | 1.5 | 2.0 | 2.5 | 3.0 | AMCR |
|------|--------|-----|-----|-----|-----|-----|-----|-----|------|
| 0.25 | Cohen | 0.67 | 0.49 | 0 | 0 | 0 | 0 | 0 | 0.470 |
| 0.25 | SmoothAdv | 0.65 | 0.55 | 0 | 0 | 0 | 0 | 0 | 0.527 |
| 0.25 | Macer | 0.68 | 0.56 | 0 | 0 | 0 | 0 | 0 | 0.541 |
| 0.25 | CONCERT | 0.68 | 0.55 | 0 | 0 | 0 | 0 | 0 | 0.535 |
| 0.5 | Cohen | 0.57 | 0.46 | 0.37 | 0.29 | 0 | 0 | 0 | 0.720 |
| 0.5 | SmoothAdv | 0.54 | 0.49 | 0.43 | 0.36 | 0 | 0 | 0 | 0.812 |
| 0.5 | Macer | 0.64 | 0.52 | 0.43 | 0.31 | 0 | 0 | 0 | 0.830 |
| 0.5 | CONCERT | 0.55 | 0.50 | 0.41 | 0.33 | 0 | 0 | 0 | 0.801 |
| 1.0 | Cohen | 0.44 | 0.38 | 0.33 | 0.26 | 0.19 | 0.15 | 0.12 | 0.863 |
| 1.0 | SmoothAdv | 0.40 | 0.38 | 0.34 | 0.31 | 0.27 | 0.25 | 0.19 | 1.001 |
| 1.0 | Macer | 0.48 | 0.43 | 0.36 | 0.30 | 0.25 | 0.18 | 0.13 | 1.004 |
| 1.0 | CONCERT | 0.45 | 0.38 | 0.35 | **0.32** | 0.21 | 0.20 | 0.13 | 0.989 |

## F    Details of Generated Variance

We observed information about the distribution of the generated $\sigma$. The statistics of per-sample $\underline{\sigma} := \min_{i \in \{1,...,d\}} \sigma_i$ calculated from 500 CIFAR10 test samples are listed in Table 9. We find most dimensions of $\sigma$ are suppressed to the vicinity of the lower bound value $a$.

Table 9: Statistics of generated $\sigma$

| $\sigma$ Range | Min | Median | max |
|-----------|-----|--------|-----|
| 0.25–0.50 | 0.2500 | 0.2503 | 0.2523 |
| 0.50–1.00 | 0.5000 | 0.5017 | 0.5302 |
| 1.00–1.50 | 1.0100 | 1.0916 | 1.1098 |

## G    Runtime and Overhead

The runtime and overhead of COLOR-CET on two Nvidia P100 GPUs are listed in Table 10.

Table 10: Runtime and Overhead

| Item | Time |
|------|------|
| Single prediction | $3.85s$ |
| Single Certification | 14.62 |
| Training epoch | $593.26s$ |

## H    Model Architecture

The details of the generator used in generating noise variance for CIFAR10 is presented below.

```
----------------------------------------------------------------
        Layer (type)              Output Shape          Param #
================================================================
           Conv2d-1            [-1, 16, 32, 32]              64
   InstanceNorm2d-2            [-1, 16, 32, 32]               0
             ReLU-3            [-1, 16, 32, 32]               0
           Conv2d-4            [-1, 32, 16, 16]           8,224
   InstanceNorm2d-5            [-1, 32, 16, 16]               0
             ReLU-6            [-1, 32, 16, 16]               0
           Conv2d-7             [-1, 64, 8, 8]          32,832
   InstanceNorm2d-8             [-1, 64, 8, 8]               0
             ReLU-9             [-1, 64, 8, 8]               0
 ReflectionPad2d-10            [-1, 64, 10, 10]               0
          Conv2d-11             [-1, 64, 8, 8]          36,864
     BatchNorm2d-12             [-1, 64, 8, 8]             128
            ReLU-13             [-1, 64, 8, 8]               0
 ReflectionPad2d-14            [-1, 64, 10, 10]               0
          Conv2d-15             [-1, 64, 8, 8]          36,864
     BatchNorm2d-16             [-1, 64, 8, 8]             128
     ResnetBlock-17             [-1, 64, 8, 8]               0
 ReflectionPad2d-18            [-1, 64, 10, 10]               0
          Conv2d-19             [-1, 64, 8, 8]          36,864
     BatchNorm2d-20             [-1, 64, 8, 8]             128
            ReLU-21             [-1, 64, 8, 8]               0
 ReflectionPad2d-22            [-1, 64, 10, 10]               0
          Conv2d-23             [-1, 64, 8, 8]          36,864
     BatchNorm2d-24             [-1, 64, 8, 8]             128
     ResnetBlock-25             [-1, 64, 8, 8]               0
 ReflectionPad2d-26            [-1, 64, 10, 10]               0
          Conv2d-27             [-1, 64, 8, 8]          36,864
     BatchNorm2d-28             [-1, 64, 8, 8]             128
            ReLU-29             [-1, 64, 8, 8]               0
 ReflectionPad2d-30            [-1, 64, 10, 10]               0
          Conv2d-31             [-1, 64, 8, 8]          36,864
     BatchNorm2d-32             [-1, 64, 8, 8]             128
     ResnetBlock-33             [-1, 64, 8, 8]               0
 ReflectionPad2d-34            [-1, 64, 10, 10]               0
          Conv2d-35             [-1, 64, 8, 8]          36,864
     BatchNorm2d-36             [-1, 64, 8, 8]             128
            ReLU-37             [-1, 64, 8, 8]               0
 ReflectionPad2d-38            [-1, 64, 10, 10]               0
          Conv2d-39             [-1, 64, 8, 8]          36,864
     BatchNorm2d-40             [-1, 64, 8, 8]             128
     ResnetBlock-41             [-1, 64, 8, 8]               0
 ConvTranspose2d-42            [-1, 32, 16, 16]          32,768
   InstanceNorm2d-43           [-1, 32, 16, 16]               0
            ReLU-44            [-1, 32, 16, 16]               0
 ConvTranspose2d-45            [-1, 16, 32, 32]           8,192
   InstanceNorm2d-46           [-1, 16, 32, 32]               0
            ReLU-47            [-1, 16, 32, 32]               0
 ConvTranspose2d-48             [-1, 3, 32, 32]              48
         Sigmoid-49             [-1, 3, 32, 32]               0
================================================================
```

Figure 7: The architecture of the generator.

