# OpenReview forum: "Concert: Context-Aware Randomized Smoothing via Colorization-Based Entropy"
_TMLR — Rejected by TMLR_

### Review · Reviewer_dndW · 2023-05-01

**Summary Of Contributions:**

This paper proposes CONCERT, which is a data-dependent randomized smoothing mechanism to enable larger certified volumes than isotropic randomized smoothing. Specifically, the authors introduce a training algorithms which optimize the data-dependent noise variance to improve the certified areas. For images, the noise is inspired by the entropy of pixels.

**Audience:**

Yes

**Broader Impact Concerns:**

As far as what I know, there is no ethical concerns about this work.

**Claims And Evidence:**

No

**Requested Changes:**

We encourage the authors to address the concerns pointed in the "weakness" part in the previous section. In addition to these questions, I have the following minor questions that may require the authors to edit the paper.

1. How Theorem 2 is related to the rest part of the paper is not clear.

2. More intuitions are needed to explan why the pixel-wise entropy is a good guess for anisotropic noise.

3. More graphic or diagrams are needed to better demonstrate the authors' idea.


**Strengths And Weaknesses:**

Strength: this paper studies data-dependent anisotropic smoothing, which is interesting and is a nice extension of the original isotropic randomized smoothing as introduced in [Cohen et al 2019].

Weakness:

1. The notation is confusing. For example, it is better to provide the mathematical formulation of $p_x(z, \sigma_1)$ and $p_{x + \delta} (z, \sigma_2)$ in Equation (4). In Lemma 3, it is confusing that $\sigma_0 = min \Sigma_0$ since $\Sigma_0 = \sigma_0 I$, same for $\sigma_1$ and $\Sigma_1$.

In Equation (6) and Equation (7), both $\xi_< (\delta, \sigma_1)$ and $\xi_> (\delta, \sigma_1)$ are both defined as  $\underline{P}_{x + \delta} (\underline{D_1})$, what are their differences?

In addition, $\Phi_{\chi^2}$ seems undefined.

2. Experiments are relatively weak. For example, in Table 1, I cannot see CONCERT's advantages over the baseline Cohen, SmoothAdv or Macer. Same for Figure 3. The difference is marginal.

3. There is one previous work with a similar motivation in ICML 2019 "On Certifying Non-Uniform Bounds against Adversarial Attacks", which also studies anisotrophic certified bound (using linear approximation instead of randomized smoothing). Is it possible that this work use the ADMM method in [ICML 2019] instead of using a regularization scheme to encourage large certified volume? Because we need to mannually set the value of the coefficient when using regularization, ADMM (maximize the volume while preserving robustness guarantee) seems to be a better choice.

---

> ### Author Response · Authors · 2023-05-22
> **Response to Reviewer dndW**
>
> We appreciate your comments and would like to thank you for the efforts you put into the review.
>  1. **Notations.**  Sorry for making confusion here. First, since the smoothing noises in each dimension are independent, $p_{x}(z,\sigma_0)$ and $p_{x+\delta}(z,\sigma_1)$ take the form of the joint PDF of independent Gaussians over $d$ input dimensions. Their formulations in our paper are as follows:
> $$
> p_{x}(z,\sigma_0) =\frac{1}{(2\pi)^{d/2}{\sigma_0}^d} \exp(-\frac{1}{2{\sigma_0}^2} \sum_{i=1}^d{(z_i-x_i)^2}),
> $$
> $$
> p_{x+\delta}(z,\sigma_1) = \frac{1}{(2\pi)^{d/2}{\sigma_1}^d} \exp(-\frac{1}{2{\sigma_1}^2} \sum_{i=1}^d{(z_i-x_i-\delta_i)^2}).
> $$
> Herein, $\sigma_0$ and $\sigma_1$ are arbitrary sample-specific standard deviations of the smoothing noise. For example, we could use $\sigma_0=\min\Sigma_0$, where $\Sigma_0$ is the generated standard deviation vector by $G$ at $x_0$. Similarly, $\sigma_1=\min\Sigma_1$, where $\Sigma_1$ is the generated standard deviation vector at $x_1$. However, we notice that $\sigma_0$ and $\sigma_1$ used in Theorem 2 should be different to the same notations used in other parts of the paper. We will accordingly remove $\sigma_0$ and $\sigma_1$ from Theorem 2 to avoid making confusion. Second, $\Phi_{{\chi}^2}$ is the CDF of the Chi-Square distribution. At last, $\xi_{<}(\delta,\sigma_1)$ and $\xi_{>}(\delta,\sigma_1)$ are different based on whether $R^2_<$ or $R^2_>$ (*i.e.,* based on whether $\sigma_0<\sigma_1$ or $\sigma_0>\sigma_1$) is used in the computation of $\underline{P}_{x+\delta}(\underline{D_1})$, which is identical to the definitions in the paper of Súkenık et al. (ICML'22).
>
> 2. **Advantages of CONCERT in the evaluation.**  Our experiments focus on the defense performance against unrestricted attacks whose perturbations are not bounded by $\ell_p$ norms. As a result, CONCERT outperforms the competitors on the adversarial testsets of cAdv and Shadow. On the other hand, CONCERT obtains comparable with other baselines on clean datasets. To make this comparison, we measure the certified accuracy of CONCERT under a very specific setting in which the minimal variance in the generated anisotropic variance matrix equals the variance used in other isotropic certification methods. Since the certified accuracy generally drops when increasing noise variance, the comparisons actually put CONCERT in worse situations than the competitors. In such comparisons,  CONCERT still obtained close or on-par performance with the competitors.
>
> 3. **ADMM.** Thanks for referring to that paper. We reckon that ADMM works towards convex optimization problems, such as the one in the ICML'19 paper. However, our objective is non-convex due to the existence of $f$ and $G$, which makes ADMM less applicable in our case. Nevertheless, we will consider similar optimization methods in our future work.
>
> 4. **Importance of Theorem 2.** First, Theorem 2 provides the definitions of $\Sigma_0$ and $\Sigma_1$ for Lemma 3. Moreover, Theorem 2 derives the decision region $D_1$ that serves in Lemma 3. Given $\underline{D_1}$ defined in Lemma 3, we can convert the problem of computing the probability measure on $D_1$ to computing the measure on $\underline{D_1}$.
>
> 5. **Why use pixel-wise entropy.** The pixel-wise entropy given by the pretrained model is a good metric for measuring the uncertainty in each pixel of the image. Henceforth, pixels with higher uncertainty can bear more perturbations without significantly reducing the naturalness of the image. We assume that unrestricted perturbations follow this rule in their perturbation allocation. The assumption is visually supported by Figure 5. To better verify this assumption, we will accordingly add a plot showing the correlation between the perturbation magnitude and the pixel-wise entropy of unrestricted perturbations crafted by different types of attacks.
>
> 6. **More graphs or diagrams.** We agree that adding more graphs to support our motivation/idea is necessary. In addition to Figure 5, we will accordingly add a figure demonstrating the relationship between the perturbation magnitude and the pixel-wise entropy of unrestricted perturbations crafted by different types of attacks. This figure will serve as a further explanation of our intuition of using pixel-wise colorization entropy to measure the dimension-wise risk towards unrestricted attacks.
>
> We have accordingly revised and uploaded the manuscript. The major changes are highlighted in the revised manuscript. We appreciate the reviewer's comments that helped us improve the paper.

---

### Review · Reviewer_V5QW · 2023-05-05

**Summary Of Contributions:**

The paper introduces and evaluates a different way of choosing the smoothing distribution for randomized smoothing defenses.

**Audience:**

Yes

**Claims And Evidence:**

No

**Requested Changes:**

The experimental evaluation should do a better job of really comparing to related prior work, since it feels like it wasn't implemented as a serious baseline. Also, I have a question about the validity of using a *function of the input* to choose the smoothing variance, though I could just be misunderstanding something.

**Strengths And Weaknesses:**

The introduction, background, and related work are thorough and situate this work well. I also think the idea, while not a major contribution, is valuable and worth sharing.

However, I feel the evaluation is not quite rigorous enough and doesn't provide a fair comparison to prior work, to the extent that I think the paper needs some revision. First of all, if there are existing methods which also choose varying amounts of noise per dimension, why are thes not included in the comparisons? Even if they are more compute intensive because they are per-sample, I think they are important to include. Also, this approach requires training a model which fits to the specific distribution on which it will be evaluated (e.g., there is a need to estimate entropy on ImageNet to use it on ImageNet). Are there really no other methods that choose smoothing noise based on the distribution as a whole? What would happen if CONCERT were used on a slightly different distribution than what the generator network was trained on?

Next, since this approach chooses different variances of smoothing, I think fixing a single smoothing noise for each experiment is not a valid comparison. To be thorough the comparison should probably include Randomized Smoothing, SmoothAdv, and MACER with multiple scales of noise.

Finally, this may be answered in the paper, but it's not immediately clear to me why it is valid to use a function G *evaluated on the possibly perturbed input $x$* to choose the smoothing variance? I say this because it seems for the guarantee to be valid, one would need to be able to know the range of possible choices of smoothing variance **for all $x'$ within the ball around $x$ with the radius that one wishes to certify.** Is this not the case? How can we use a black box function of the input itself to choose the variance and still provide a valid robustness certificate? Please let me know if I've missed the explanation for this.

---

> ### Author Response · Authors · 2023-05-22
> **Response to Reviewer V5QW**
>
> Thank you for your comments. We would like to address the comments as follows.
> 1. **Comparison with other input-dependent certifications.** We agree that adding more comparisons with other input-dependent randomized smoothing methods would be helpful. However, existing input-dependent certification methods have defects that lead to mathematically invalid certification. The problem is also mentioned in a concurrent work (IDRS [1*]) with ours. Therefore, we did not make comparisons with other pre-existing methods. On the other hand, IDRS is a concurrent work and has not published their code yet, it makes the comparison with IDRS less practical.
>
> [1*] Peter Súkenık, Aleksei Kuvshinov, and Stephan Günnemann. Intriguing properties of input-dependent randomized smoothing. In International Conference on Machine Learning, pp. 20697–20743. PMLR, 2022.
>
> 2. **CONCERT on a different distribution.** In the evaluations, the generator $G$ is trained on the training partition of the employed datasets and is evaluated on data from the corresponding testing datasets. The results demonstrate that $G$ can be successfully generalized to the testing datasets. As for cross-dataset generalization, we agree it would be an interesting point to study. However, it is out of the scope of this paper. We make further investigate this point in our future work.
>
> 3. **Noise scales in comparison.** We set fixed standard deviation values (*i.e.,* $0.25$, $0.5$, and $1$) for the competitors and use these values as the lower bounds (*i.e.,* $a$) of the variance generated by Concert generator (as indicated in the Metrics section on Page 8 and Table 1). Therefore, we make sure in each comparison, the certified accuracy of CONCERT is obtained under a smoothing noise that is greater or at least equal to that of the competitors. Since the certified accuracy generally drops when increasing noise variance, the comparisons actually put CONCERT in worse situations than the competitors. In such comparisons,  CONCERT still obtained close or on-par performance with the competitors.
>
> 4. **$G$ on perturbed $x$.** Thanks for pointing this out. It is possible that $G$ may be evaluated on perturbed $x'$ to determine the smoothing variance. However, we do not need to know all possible choices for the smoothing variance since the variance generation step is independent of the certification process. Given an input $x$ (or $x'$), its smoothing noise is parameterized by $G(x)$ (or $G(x')$).  Next, the certification makes sure there is no adversary in the vicinity of $x$ (or $x'$) that changes its prediction by the classifier. This is in consistent with the aim of randomized smoothing, which only ensures the expected prediction will not be changed but has no control over the correctness of the original prediction.
>
> We have accordingly revised and uploaded the manuscript. The major changes are highlighted in the revised manuscript. We want to thank you again for the effort you put into the reviewing process!

---

### Review · Reviewer_pedU · 2023-05-10

**Summary Of Contributions:**

This paper proposes a variant of randomized smoothing which smooths images asymmetrically based on the (estimated) pixel-wise entropy, rather than smoothing using an isotropic Gaussian. The resulting technique performs similarly to existing baselines on lp-constrained adversarial attacks, and better performance against unrestricted "semantic attacks."

**Audience:**

Yes

**Claims And Evidence:**

No

**Requested Changes:**

Please see the section above. In summary:

1. Significant proofreading
2. Precisely defining the threat model and evaluating within that threat model
3. Using adaptive attacks targeted towards the defense

**Strengths And Weaknesses:**

Strengths:

1. The paper studies an interesting problem (data-dependent robustness guarantees rather than uniform $\ell_p$ guarantees) and proposes a novel solution concept.
2. The evaluation is extensive, and the authors compare against the appropriate baselines (SmoothAdv, RS, and Macer).
3. The authors theoretically or intuitively justify each design decision, and study the effect of hyperparameter choices on their method's performance.

That said, I think the paper needs significant changes before I can recommend its publication. My main questions/concerns are below:

1. The writing of the paper is hard to follow and made the paper very difficult to read. I tried not to let this affect my evaluation of the paper, but it's possible that I missed important details. The authors should be careful to define terms like "tightness of a robustness region," "quantifying the perturbation space," "dimension-agnostic," etc. and the paper should also be proofread to remove many run-on/unclear sentences.

2. On the experimental front, I think the paper should do a better job of defining the precise threat model of interest. It is okay if this threat model is not that of lp-bounded attacks (e.g., maybe the threat model is an l2-bounded attack in the representation space of a neural network, or an l2-bounded attack but only within certain parts of the image), but as it stands now the paper does not characterize the restrictions placed on the adversary and so it's hard to contextualize the results. It also makes proper evaluation nearly impossible (see point 4), as the authors are forced to use precomputed "unrestricted" adversarial examples to measure robustness.

3. Similarly, I'm a bit confused about the setup; based on Figure 2, it seems like the smoothing parameter generator is run at test time. Given that this generator is a neural network, why can't the adversary attack this neural network and force it to output unreasonable parameters? Perhaps this would not change the $\ell_p$-certified results, but it seems like this could break the other "unrestricted" attacks.

4. In general, evaluation should not be performed against pre-computed adversarial examples for the base classifier, as such examples are often simply non-transferable or non-robust to noise. It's thus hard to draw conclusions from the results on CAdv and the like. It is important to only use adversarial perturbations that were generated adaptively for the defense itself.

---

> ### Author Response · Authors · 2023-05-22
> **Response to the comments by Reviewer pedU**
>
> Thanks to the reviewer for the very helpful comments. We would like to address the comments point by point.
>
> 1. **Proofreading.** We will accordingly revise the paper to remove or revise ambiguous terms/sentences.
>
> 2. **Threat model (restrictions placed on perturbations).** It is hard to place actual restrictions on unrestricted attacks. Instead, in our experiment, we measure the certified accuracy of unrestricted attacks based on subsets of unrestricted examples characterized by the maximal $\ell_2$ norm of their perturbations (i.e., $S=\\{\|x’-x\|_2 \leq \tau\\}$ for different $\tau$ values). In this way, we can make comparisons on sets of unrestricted perturbations bounded by certain $\ell_2$ norms. In addition, we add a new section to state the problem definition and explain our motivation for using pixel-wise entropy in deciding the variance of the smoothing noise.
>
> 3. **Adaptive attacks.** The adaptive attack against the defense could be intriguing. It should be noticed that there is no adaptive attack against the certified defense itself within the robustness radius of an input. However, possible adaptive attacks may exist against the generator $G$. To address the concern of the reviewer, we will add an adaptive attack that manipulates the generated variance values to shrink the certified robust region around $x$ such that the perturbed $x'$ can escape the robust region. The optimization of such $x'$ is simple:
> $$
> \delta^* = argmin_{\delta} (L_{adv} + \lambda \vert|G(x+\delta)\vert|),
> $$
> where $L_{adv}$ is the vanilla adversarial loss function for the cAdv/Shadow attack, $\vert|\cdot\vert|$ is the $\ell_2$ norm, and $\lambda$ is a constant. Nevertheless, we would like to clarify that this attack assumes the attacker knows the existence of the defense and can directly access the output and the gradients of $G$. This is a strong assumption in many cases in which the attacker only has access to querying and getting final predictions from a black-box model in an end-to-end manner. Moreover, due to that the output of $G$ is bounded between $[a, \omega b]$, setting $a$ to a proper value can effectively mitigate such an adaptive attack.
>
> 4. **Evaluation.** The reason why we evaluated CONCERT on both the cAdv examples and the Shadow examples is to show that the anisotropic robust regions induced by CONCERT can better align with the unrestricted perturbation space. In other words, perturbations that have a large magnitude in certain dimensions can still be captured by the anisotropic robust regions, such that the predicted labels from the smoothed classifier are not changed. On the other hand, as mentioned above, we will consider an adaptive attack against the generator $G$ in our revised manuscript.
>
> We have accordingly revised and uploaded the manuscript. The major changes are highlighted in the revised manuscript. We also had a proofread to eliminate ambiguous words/sentences and typos. We would like to express our appreciation to the reviewer for the comments and further feedback.

---

### Decision · Action_Editors · 2023-06-09

**Recommendation:** Reject

**Comment:**

All three reviewers recommended against acceptance, due to not agreeing with the following criteria for acceptance: "Are the claims made in the submission supported by accurate, convincing and clear evidence?" I thus follow their recommendations.

**Audience:**

The results are topical and interesting to the TMLR audience.

**Claims And Evidence:**

The primary critique that all three reviewers had was the lack of a precise threat model. This makes it difficult to ascertain what the formal claim actually is, and consequently whether evidence is provided to support it.

Reviewer VQ5W still had concerns about the non-adaptive attacks on the input-dependent choice of dimensionwise variance made by G.

Reviewers also said that the writing needed to be improved in terms of notation and motivation.

They overall felt that this work is missing real evidence of the claims presented.

**Resubmission Of Major Revision:**

The authors may consider submitting a major revision at a later time.